# Climate co-benefits of tiger conservation

Aakash Lamba [1,2] ✉, Hoong Chen Teo [1,2], Rachakonda Sreekar[1,2], Yiwen Zeng [1,2,3,4], Luis Roman Carrasco [1,2] & Lian Pin Koh [1,2,4] ✉

Biodiversity conservation is increasingly being recognized as an important co-benefit in climate change mitigation programmes that use nature-based climate solutions. However, the climate co-benefits of biodiversity conservation interventions, such as habitat protection and restoration, remain understudied. Here we estimate the forest carbon storage co-benefits of a national policy intervention for tiger (*Panthera tigris*) conservation in India. We used a synthetic control approach to model avoided forest loss and associated carbon emissions reductions in protected areas that underwent enhanced protection for tiger conservation. Over a third of the analysed reserves showed significant but mixed effects, where 24% of all reserves successfully reduced the rate of deforestation and the remaining 9% reported higher-than-expected forest loss. The policy had a net positive benefit with over 5,802 hectares of averted forest loss, corresponding to avoided emissions of $1.08 \pm 0.51$ MtCO$_2$ equivalent between 2007 and 2020. This translated to US\$92.55 $\pm$ 43.56 million in ecosystem services from the avoided social cost of emissions and potential revenue of US\$6.24 $\pm$ 2.94 million in carbon offsets. Our findings offer an approach to quantitatively track the carbon sequestration co-benefits of a species conservation strategy and thus help align the objectives of climate action and biodiversity conservation.

Biodiversity conservation and climate change mitigation are intimately linked, but have historically been addressed as separate challenges[1]. However, there is an urgent need to align both goals and synergize resource allocation for achieving these objectives given the rapid pace of global biodiversity declines and the rising impacts of climate change[2,3]. Nature-based solutions, particularly through habitat protection and restoration, are an important approach that can help accomplish both goals simultaneously[4]. Biodiversity conservation is recognized and valued as a key co-benefit of climate change mitigation projects that implement nature-based climate solutions. For example, forest carbon offsetting projects that integrate biodiversity co-benefits into their stated goals perform substantially better in terms of market preference compared with those projects that focus only on carbon reductions[5,6].

At the same time, land-management interventions with the express goal of conserving biodiversity could provide ancillary climate change mitigation benefits. This juxtaposition of biodiversity conservation as the primary benefit, instead of climate change mitigation, represents an important paradigm for the preservation of natural carbon stocks[7–9]. We argue that a biodiversity-first approach, which helps quantify the downstream benefits of biodiversity preservation on climate mitigation targets, can help further incentivize species conservation programmes while achieving climate action through the avoided social cost of the loss of natural ecosystems.

Furthermore, this paradigm potentially unlocks unforeseen opportunities for funding conservation programmes using financial instruments, such as carbon offsets[8], which have been growing immensely as a source of funding for nature-based climate solutions[10]. In theory, protected areas are gazetted for biodiversity conservation, rendering them not additional[11], and hence not eligible for carbon credits. In practice, however, some protected areas across the tropics

[1]Centre for Nature-based Climate Solutions, National University of Singapore, Singapore, Singapore. [2]Department of Biological Sciences, National University of Singapore, Singapore, Singapore. [3]School of Public and International Affairs, Princeton University, Princeton, NJ, USA. [4]Tropical Marine Science Institute, National University of Singapore, Singapore, Singapore. ✉e-mail: aakash.lamba@u.nus.edu; lianpinkoh@nus.edu.sg

are also known to function as 'paper parks', with continued degradation within boundaries often conferring limited biodiversity conservation benefits[12]. This can stem from a variety of reasons, including a shortfall in funding[13]. Revenues from the trade of fungible carbon offsets, which represent standardized and internationally tradable reductions in emissions, arising from the recognition of the climate change mitigation benefits associated with biodiversity conservation can serve as a means of closing this funding gap[14].

To demonstrate this approach, we evaluated the forest carbon storage co-benefits of a nationwide policy intervention for a species conservation programme, specifically the conservation of the tiger (*Panthera tigris*) in India. The tiger is one of the most charismatic and highly protected wild species in India. As the range country with the highest proportion of the world's wild tigers, India's conservation policies are crucial to the long-term survival of tigers and the persistence of their associated habitats, which provide a suite of economic and sociocultural services[15]. Although Project Tiger was launched in 1973 to save India's wild tigers from the brink of extinction, the National Tiger Conservation Authority (NTCA) was established in 2005 to oversee and further improve the national tiger conservation strategy in India (https://www.ntca.gov.in/). As part of this initiative, key protected areas were designated as tiger reserves in India. This gazette notification entails enhanced management through better protection, monitoring and funding for these protected areas[16]. As of 2022, over 52 reserves have undergone this additional level of conservation management in India (https://www.ntca.gov.in/).

Although preserving and increasing tiger populations is the primary objective of this policy, it could also provide enhanced benefits to forest protection and associated carbon emission reductions through avoided deforestation. Reserves that fall under this policy must prepare a Tiger Conservation Plan, which includes measures for regulating the extraction of forest products, reducing deforestation drivers and encouraging alternative livelihoods for communities that live within tiger conservation landscapes[17]. We hypothesized that the implementation of this policy and the resultant improved protected area management, particularly through better enforcement, would lead to a decrease in forest loss in affected protected areas due to a reduction in deforestation drivers.

To model the effects of these enhanced conservation measures on forest loss rates, we applied a synthetic control approach, which allows the simulation of counterfactual baseline deforestation trajectories in protected areas that underwent this policy intervention (hereon referred to as tiger reserves). This causal inference method matches the response variable, in this case, cumulative forest loss of the 'treatment' group (reserves with tiger reserve status) to a weighted model of untreated protected areas, or a 'donor' pool (protected areas with known tiger presence but not designated as tiger reserves by the NTCA), before the policy intervention took place[18] (see Fig. 1 for locations of tiger reserves and untreated protected areas). To model the underlying structural drivers of forest cover loss we included reserve-level variables associated with deforestation, which included anthropogenic disturbances, history of protection, poverty indices, geographical attributes, forest quality at the start of the study period and climatic variables as covariates in the matching process (see Methods for details on covariates for drivers of forest loss)[19–24]. We extrapolated these synthetic counterfactual models to the period after the implementation of the conservation policy to evaluate the difference between the cumulative forest loss in tiger reserves and their respective synthetic counterfactual models. We translated the forest change due to the designation of tiger reserves to equivalent $CO_2$ ($CO_2e$) emissions averted using average aboveground and belowground biomass carbon densities for each reserve for the year the intervention was implemented[25]. We consequently converted these emissions into an ecosystem services value based on the avoided social cost of carbon emissions in India and potential revenue from carbon offsets in the voluntary carbon market, which represents one of the most well-established mechanisms for payment for ecosystem services (see Methods for details)[10,26,27].

## Results

### Forest loss across reserves

For the period 2001–2020, the total forest loss in 162 protected areas with tiger presence, which includes both treated and untreated reserves used in the analyses, was 61,648 ha or 3,082 ha of loss per year. Over 77% of these losses (47,719 ha) were from untreated reserves ($n = 117$), with a mean [95% confidence interval] cumulative forest loss of 408 [190–626] ha per reserve. Protected areas designated as Tiger Reserves ($n = 45$) contributed to 23% of the total forest loss in the study period (13,289 ha), with a mean [95% confidence interval] cumulative forest loss of 309 [127–496] ha per reserve. More than half of the treated reserves (51%) underwent treatment in 2007, whereas 2015 was the most recent intervention year used in the analyses. Regarding the performance of protected areas, the highest observed deforestation occurred in the Kotgarh Wildlife Sanctuary in the state of Odisha, which lost over 8,927 ha of forest (28% of forest area since 2000). Bor Wildlife Sanctuary in Maharashtra was the only treatment reserve that did not exhibit any deforestation between 2001 and 2020.

### Synthetic control analysis

Of the 45 tiger reserves that underwent the conservation policy intervention, 15 showed significant but mixed results ($P < 0.05$) after reserves with anticipation effects were excluded from analyses (only Pench Tiger Reserve; Supplementary Table 3 and Supplementary Figs. 6 and 7). Reserves that showed a significant effect of the tiger conservation policy on deforestation represented a net avoided forest loss value of 5,802 ha. Eleven of these 15 tiger reserves exhibited significant avoided deforestation of 6,558 ha since 2007 (Fig. 2 and Extended Data Fig. 1). Nawegaon–Nagzira Tiger Reserve showed the highest averted forest loss (2,645 ha) since its notification as a tiger reserve in 2013. However, four reserves exhibited increased forest loss despite the intervention, where the observed cumulative forest loss trajectories were significantly higher than the modelled counterfactuals. These reserves experienced 756 ha of additional forest loss compared with their counterfactuals (Fig. 3). Pilibhit Tiger Reserve lost the highest forest compared with its synthetic counterfactual, with over 300 ha of additional forest lost since its treatment year in 2008. No tiger reserve in the Northeast Hills and Brahmaputra regions avoided deforestation.

There was no statistical difference between the mean square prediction errors (MSPE) in the pre-intervention period between reserves that showed significant results ($n = 15$) and those with insignificant results ($n = 30$) (bootstrap hypothesis testing, $P = 0.21$), indicating a comparable quality of fit. Similarly, there was no statistical difference in pre-intervention MSPE values in northeastern tiger reserves compared with those in other regions (bootstrap hypothesis testing, $P = 0.49$). Moreover, the synthetic controls visually matched the observations in the pre-intervention period for both groups. The distribution of variables associated with the drivers of forest loss between donor and treatment reserves was comparable, suggesting no significant structural difference between both groups (Supplementary Fig. 3). In addition, our results were robust to area-based trimming of donor pools (see robustness checks in Methods). Effects of the intervention with trimmed donor pools exhibited the same direction for all modelled scenarios; that is, whether the effect was consistent in terms of avoided loss or increased loss (Supplementary Table 4). In addition, 73% (11 of 15) and 75% (6 of 8) of the results with a trimmed donor pool of 10% and 25%, respectively, were within our ±20% final findings. Similarly, over 87% (13 of 15) and 62% (5 of 8) of the results still exhibited significant effects based on placebo tests (Supplementary Table 4).

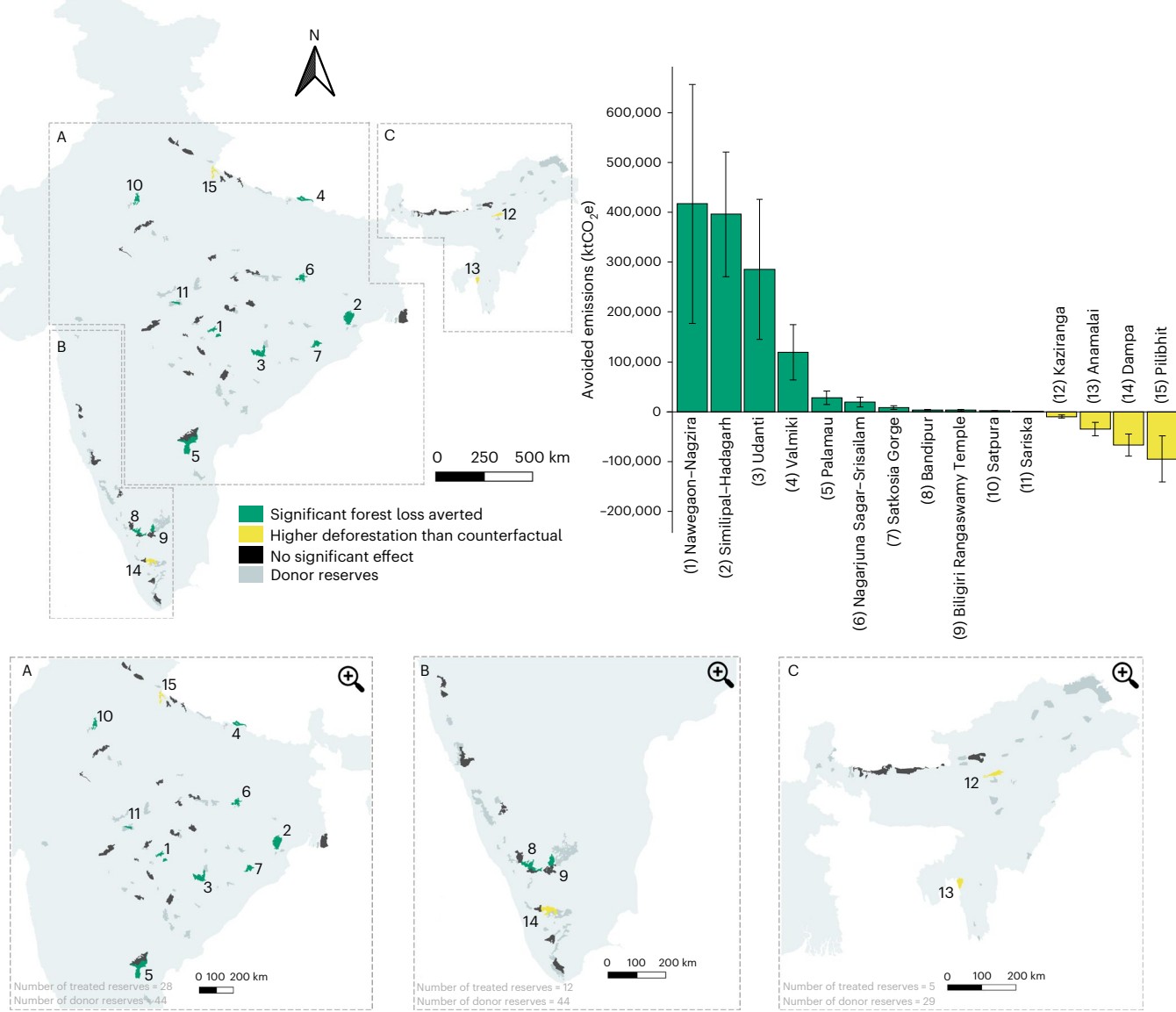

**Fig. 1 | The locations and avoided carbon emissions of analysed tiger reserves.** Top left: map of India with the reserves analysed in the study (boundaries from OSM[43]). Light grey donor reserves are protected areas with tiger presence that did not undergo enhanced conservation policy and were used to generate synthetic counterfactuals for tiger reserves. Dotted bounded boxes represent geographical areas used for donor matching. To ensure robust significance testing, more than 20 donor reserves or placebos were needed per geographical grouping. Therefore, contiguous tiger conservation landscapes[54] were combined into the following groups: Shivalik–Gangetic, Central India, Eastern Ghats and Sunderbans regions (A); Western Ghats (B); and Northeast Hills and Brahmaputra region (C). For significance testing, we used a two-sided Fisher's exact test to compare the ratios of pre-intervention and post-intervention mean squared prediction errors between the treated synthetic reserve and placebo units for each tiger reserve (see Extended Data Fig. 1 for unadjusted $P$ values of tiger reserve with significant effects). Green reserves represent tiger reserves that exhibited significantly avoided deforestation while yellow reserves are treated reserves where the observed deforestation was significantly higher than the synthetic counterfactual ($P < 0.05$). Dark grey reserves represent tiger reserves that yielded insignificant results (see Supplementary Table 2 for list of $P$ values). Top right: total avoided emissions per reserve. Data are presented as mean avoided emissions ± uncertainty values (based on mean cumulative standard errors reported by ref. 25) in ktCO$_2$e. Error bars were derived by multiplying avoided deforestation, an emissions factor of 3.67 and mean values of the cumulative standard errors in predictions of above- and belowground carbon biomass densities reported by ref. 25 for each reserve. Reserves are numbered and colour coded to indicate locations on maps. Bottom, zoomed-in geographical zones with the number of tiger reserves and donor reserves used for deriving counterfactuals and statistical significance included for each zone. Map boundaries from OpenStreetMap[43] under a Creative Commons license CC BY-SA 2.0.

## Averted emissions and associated value

The net avoided emissions from the 15 reserves that exhibited significant results corresponded to 1.08 ± 0.51 MtCO$_2$e (see Fig. 1 for avoided emissions for each reserve). Of these avoided emissions, 0.85 ± 0.33 and 0.23 ± 0.18 MtCO$_2$e came from aboveground and belowground carbon stocks, respectively. This translated to ecosystem service provisioning of US$92.55 ± 43.56 million based on a US$86 per ton social cost of carbon estimate in India[26]. In addition, these avoided emissions corresponded to a total carbon offset value of US$6.24 ± 2.94 million based on US$5.8 per ton of CO$_2$e, the average price of carbon in the voluntary carbon market[10]. Avoided emissions from reserves that exhibited avoided deforestation ($n = 11$) was 1.28 ± 0.59 MtCO$_2$e. At a social cost of carbon value of US$86 per ton of CO$_2$e (ref. 26), these reserves provided ecosystem services through climate change mitigation of US$110.29 ± 50.87 million and represent US$7.44 ± 3.43 million (Table 1). The three most valuable reserves in

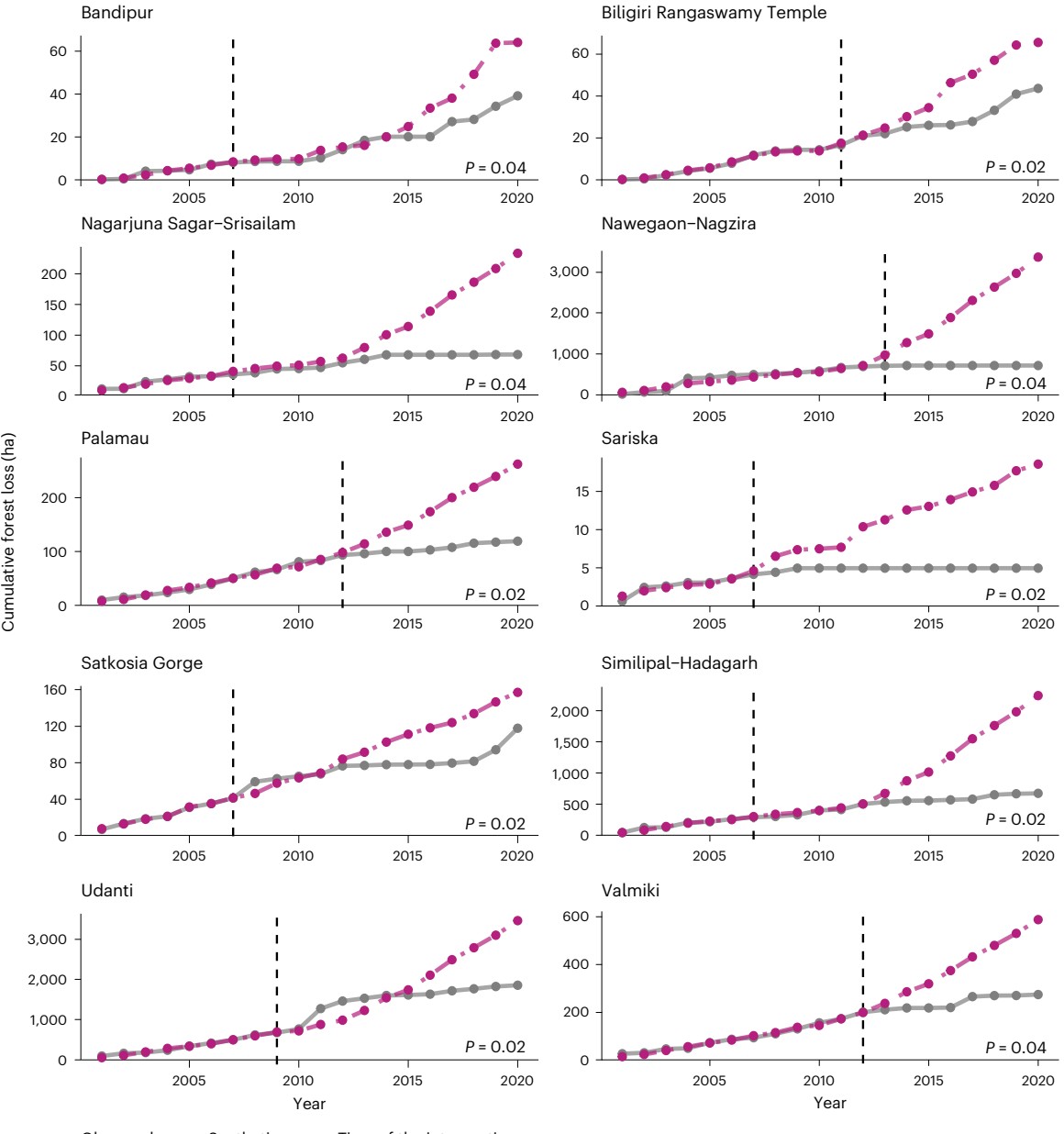

**Fig. 2 | Trend lines for cumulative forest loss in tiger reserves that exhibited significant avoided deforestation.** All reserves displayed here exhibited significant avoided deforestation (unadjusted $P < 0.05$) based on a two-sided Fisher's exact test to compare the ratios of pre-intervention and post-intervention mean squared prediction errors between placebo and treated units (see Methods for details). Significance levels (unadjusted $P$ values) were reported for each synthetic counterfactual in the displayed plots. Only reserves that exhibited avoided forest loss of more than 10 ha are displayed here (see Extended Data Fig. 1 for trend lines for all 11 tiger reserves with significant avoided forest loss in the study period). The dotted pink line represents the cumulative forest loss for the synthetic control model, whereas the dotted grey line represents observed deforestation in hectares. The vertical dashed line represents the year of implementation of the enhanced conservation policy. For each of these reserves, the synthetic control line closely tracks the observed cumulative forest loss values before the intervention.

terms of ecosystem service valuation were Nawegaon–Nagzira in Maharashtra, Similipal–Hadagarh in Orissa and Udanti–Sitanandi in Chhattisgarh. Reserves that showed higher deforestation than their synthetic counterfactuals ($n = 4$) corresponded to additional emissions of $0.21 \pm 0.09$ MtCO$_2$e, which equate to US\$17.74 ± 7.32 million in damages due to the social cost of carbon emissions and over US\$1.2 ± 0.49 million loss in potential carbon revenue in the voluntary carbon market. The three worst performing reserves in terms of additional emissions were Pilibhit in Uttar Pradesh, Anamalai in Tamil Nadu and Dampa in Mizoram (Figs. 1 and 3).

## Discussion
Our study provides a comprehensive appraisal of the national tiger conservation policy in India, demonstrating there are important ancillary climate co-benefits of enhanced protection in tiger reserves. Using a robust causal inference methodology, we could effectivity attribute avoided forest loss and consequently avoided emissions to a species-focused conservation intervention. Therefore, our findings offer empirical evidence at a broad geographical scale to support a biodiversity-first approach to climate change mitigation.

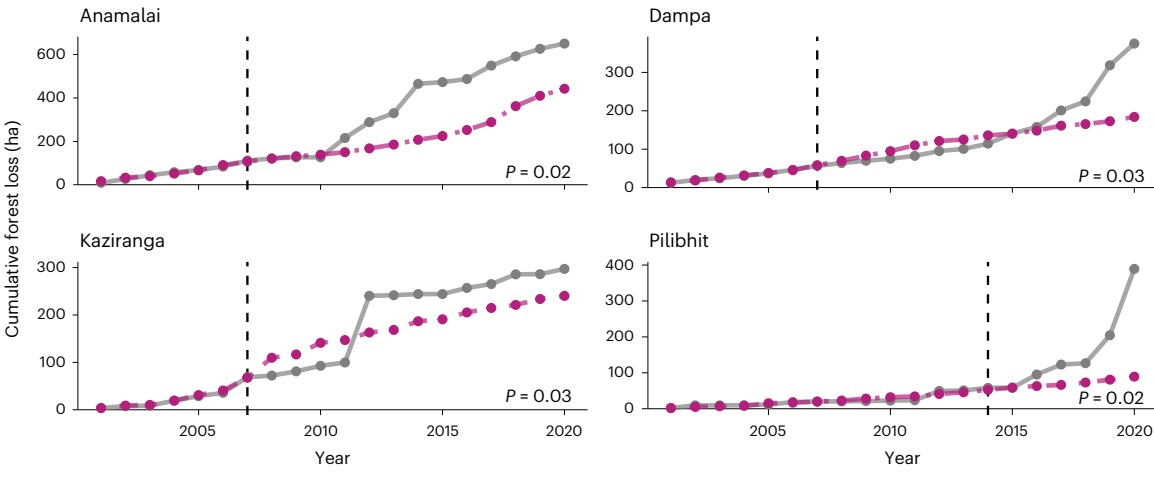

**Fig. 3 | Trend lines for cumulative forest loss in tiger reserves that exhibited significantly higher than anticipated deforestation.** All reserves displayed here exhibited significant results (unadjusted $P < 0.05$) based on a two-sided Fisher's exact test to compare the ratios of pre-intervention and post-intervention mean squared prediction errors between placebo and treated units (see Methods for details). Note the observed deforestation values were higher than the synthetic counterfactual in the case of these reserves, suggesting higher than anticipated forest loss. Significance levels (unadjusted $P$ values) were reported for each synthetic counterfactual in the displayed plots. The dotted pink line represents the cumulative forest loss for the synthetic control model, whereas the dotted grey line represents observed deforestation in hectares. The vertical dashed line represents the year of implementation of the enhanced conservation policy. For each of these reserves, the synthetic control line closely tracks the observed cumulative forest loss values before the intervention.

Overall, the designation of tiger reserves had a net positive impact on forest protection, given the significant avoidance of forest loss in a majority of these reserves (11 of 15 significant results) during the study period. In general, reserves with higher starting forest cover tended to have higher change over time, with the most positive change associated with reserves in Central India. The observed avoided deforestation in these reserves is an important outcome of India's tiger conservation policy because many of these protected areas are highly beneficial to ensuring connectivity in tiger habitats. For example, Nawegaon–Nagzira, which was the best-performing reserve in our analyses, has a vital role in ensuring landscape connectivity in tiger habitats in Central India[28].

Benefits in habitat protection through avoided forest loss probably stem from three key developments in tiger conservation strategy. First, the NTCA enacted better mechanisms for managing funds, which prevented delays in the disbursement of resources through the creation of local fund repositories in the form of tiger conservation foundations[16]. Next, conservation benefit sharing, particularly from ecotourism revenues, with communities was probably pivotal in reducing local pressures on forest areas[16]. Finally, the adoption of enhanced conservation monitoring technologies, especially GPS-based mobile tools, would have probably helped park managers in ensuring that forest guards patrolled protected areas more effectively[16].

However, for some tiger reserves, the rate of forest loss was higher than expected compared with their synthetic counterfactual. Half of these reserves (two of four) fall in Northeast India. We postulate that this may have been caused by the prevalence of reserve-specific deforestation drivers, such as encroachment, shifting agricultural practices, illegal timber trade and mining, which have historically been reported in the peripheries of notable tiger reserves, such as Kaziranga and Dampa[29,30]. Moreover, the remoteness and lower development of reserves in northeastern India have probably led to less effective enforcement and a higher risk of deforestation[31]. Therefore, our study highlights important priority areas for managers to mitigate local threats and, consequently, enhance habitat protection measures in these reserves. Overall, there were no reserves in northeastern India that had any avoided deforestation, despite reliable model fits, which indicates the conservation policies were plausibly inadequate in countering the intrinsically high rates of deforestation in the region[24].

The other two reserves that underperformed, compared with their synthetic counterfactuals, were Anamalai Tiger Reserve in Tamil Nadu and Pilibhit Tiger Reserve in Uttar Pradesh. Anamalai appears to have had a spike in deforestation starting in 2010, with over 80% of the total forest loss occurring after 2010. Extensive monoculture plantations within the region may be a driver for increased forest fragmentation[32]. The high rates of deforestation in Pilibhit could be explained by the fact that the reserve was notified more recently, in 2014. The age of a protected area is an important factor negatively correlated with forest loss[33]. Despite the higher-than-expected forest loss in Pilibhit, however, the tiger populations have been steadily growing in the reserve, with the reserve having more than doubled its population of wild tigers since its establishment[34].

This population growth despite forest loss suggests that addressing human–wildlife conflict may have been a more important determinant of tiger numbers in Pilibhit compared with the extent of available habitat[16]. It is unclear whether the prevalence of conflict is correlated with the performance of a tiger reserve in avoiding forest loss, but it is conceivable that there is an association because reserves showing no effect on avoiding forest loss, such as Nagarhole, have previously reported high costs of human–tiger conflict from livestock damage and human casualties[35]. Therefore, investigating the relationship between conflict and deforestation should be an important priority for future research because, to our knowledge, no publicly available dataset has comprehensive information on damages from human–tiger conflict across all the tiger reserves in India.

However, despite some reserves experiencing a higher-than-expected forest loss, the designation of tiger reserves in India had a net positive climate benefit. A combined estimate, which included both the reserves that experienced carbon stock loss and the high-performing reserves that averted forest loss, yielded approximately US$93 million in ecosystem services from the avoided social costs of emissions. Although the avoided emissions from the intervention, approximately 0.08 $MtCO_2e$ per year, are small compared with India's annual emissions and its nationally determined contributions as part of the

**Table 1 | Avoided deforestation and associated climate co-benefits of tiger reserves**

| Tiger reserve | Treatment year | Avoided forest loss (ha) | Avoided emissions (ktCO₂e) | Avoided social cost (US$1,000) | Carbon offset value (US$1,000) |
|---|---|---|---|---|---|
| Nawegaon–Nagzira | 2013 | 2,645 | 416.95±239.67 | 35,858.08±20,611.8 | 2,418.34±1,390.1 |
| Similipal–Hadagarh | 2007 | 1,570 | 395.9±124.66 | 34,047.31±10,720.65 | 2,296.21±723.02 |
| Udanti | 2009 | 1,611 | 285.78±140.68 | 24,577.02±12,098.34 | 1,657.52±815.93 |
| Valmiki | 2012 | 315 | 119.56±55.51 | 10,282.23±4,773.6 | 693.45±321.94 |
| Nagarjuna Sagar–Srisailam | 2007 | 167 | 19.09±10.14 | 1,641.99±872.11 | 110.74±58.82 |
| Palamau | 2012 | 143 | 28.01±13.34 | 2,409.03±1,147.02 | 162.47±77.36 |
| Satkosia Gorge | 2007 | 39 | 8.09±3.42 | 695.52±294.01 | 46.91±19.83 |
| Bandipur | 2007 | 25 | 2.94±1.58 | 252.41±135.93 | 17.02±9.17 |
| Biligiri Rangaswamy Temple | 2011 | 22 | 3.68±1.35 | 316.86±115.97 | 21.37±7.82 |
| Sariska | 2007 | 14 | 0.68±0.45 | 58.21±38.36 | 3.93±2.59 |
| Satpura | 2007 | 9 | 1.75±0.76 | 150.5±64.93 | 10.15±4.38 |

Total avoided forest loss in hectares, avoided emissions in ktCO₂e, the value of ecosystem services provided in units of $US1,000 at a social cost of carbon estimate of US$86 per ton of CO₂e, and a potential carbon market revenue based on US$5.8 per ton CO₂e. Values are listed for the 11 reserves out of the 15 significant results that exhibited avoided deforestation.

Paris Agreement, this is nonetheless an important contribution given that India ranks as the country that is most vulnerable to the impacts of climate change in terms of the social cost of carbon, with each additional ton of emissions leading to a loss of US$86 to the Indian economy[26].

Moreover, the budget for Project Tiger in 2020–2021 was just under US$27 million based on 2020 conversion rates (www.moef. gov.in/). More than a quarter of this budget was paid back in over US$7 million per year between 2007 and 2020 from the avoided social cost of emissions. Although these annual returns are a fraction of the annual management costs of these reserves, they demonstrate that resources invested in biodiversity conservation can reimbursed in the form of economic benefits from ecosystem services. In addition, had these enhanced protection measures been enacted in the untreated protected areas with tiger presence, an additional US$38 million could have been gained from ecosystem services due to avoided emissions (based on a 0.21% rate of forest loss avoidance per hectare of forest cover since the baseline year across all analysed tiger reserves). These estimates provide a realistic image of the scale and associated timeframes associated with the climate co-benefits of biodiversity conservation to key stakeholders such as communities, researchers and policymakers.

Apart from the avoided social costs from emissions, carbon markets could be an important means of realizing the value associated with the ecosystem services provided by biodiversity conservation initiatives. Carbon markets are one of the most well-established and popular payment systems for ecosystem services[27]. The voluntary carbon market has grown rapidly in the recent past, with nature-based climate solutions, particularly through forest protection and restoration, emerging as one of the most rapidly growing types of carbon offset[10]. Here we estimate that the avoided emissions from the tiger conservation policy in India corresponded to more than US$6 million between 2007 and 2020.

However, major barriers to mobilizing such forms of conservation funding remain. Additionality, or the requirement that an intervention must provide additional climate benefits compared with a business-as-usual scenario (without the intervention), is a necessary and vital component of carbon markets[36]. The implementation of the national tiger conservation strategy does not necessarily meet the current requirements of additionality in its current form[11]. Tiger reserves were already protected before the enhanced conservation policy and are, therefore, technically ineligible for receiving funds from carbon markets because they do not meet the fundamental criterion of additionality. This implies that the US$6.2 million saving will probably remain untapped through existing avoided emissions methodologies. A plausible alternative could be to recognize the potential for improved management of protected areas to provide additional carbon benefits. Regardless of the strategy used, realizing this value could meet a substantial proportion of the financial costs associated with safeguarding individual reserves.

There is an urgent need for additional financial resources to close the funding gap for species conservation[13], and we show that there are clear synergies between such biodiversity conservation policies and carbon markets. This represents a missed opportunity, where adjustments to the carbon markets could divert money to fund conservation and benefit local communities. However, given the historical costs of conservation policies borne by local communities around tiger reserves, particularly through displacement[37,38], it is crucial that future attempts to integrate tiger conservation policy into carbon markets be cognizant of the cultural, social and economic needs of local communities, whose participation is crucial in equitable and effective conservation[39].

We acknowledge some caveats in our study. First, the carbon benefits of this approach apply primarily to species found in high-carbon ecosystems. However, given the large overlap of the world's high biodiversity areas with carbon-rich protected areas, it is all the more urgent that such evaluations be carried out to create an evidence base for channelling more resources into the protection of such reserves[40]. Second, we relied on protected area boundaries obtained from OpenStreetMap (OSM), which may introduce uncertainties in our analyses given that these maps are created by publicly available, user-generated data. This is, to our knowledge, the most updated and representative publicly available dataset—for instance, the most recent version of the World Database on Protected Areas includes shapefiles of less than 5% of India's published reserves[41]. Moreover, our findings mark an important first step in recognizing the potential scale of climate benefits of a biodiversity-first approach, which we hope that protected area managers can further build on using our framework along with high-resolution, on-ground data to aid more effective habitat protection. Lastly, our study offers only a conservative estimate of avoided carbon emissions because we focus on only forest loss, whereas degradation may be an important driver of carbon loss in terrestrial forests[42]. The development of a regional forest degradation dataset, which spans the varied forest types represented in India's tiger reserves, is a pressing priority for future research.

Despite these limitations, however, our findings demonstrate that there are important ancillary climate co-benefits of enhanced biodiversity protection measures through additional avoided

deforestation in carbon-rich ecosystems, such as terrestrial forests. Using a robust causal inference methodology, our study provides a framework to show climate change mitigation additionality of enhanced conservation within protected areas. Our findings suggest that integrating species conservation programmes into global carbon markets can help unlock additional opportunities for funding the protection and restoration of natural habitats. Finally, our study reiterates that biodiversity conservation is at the core of climate change mitigation and underscores the need for aligning the goals of biodiversity preservation and climate impact.

## Methods

### Compilation of protected areas

We extracted protected area boundaries from OSM, which is an open-source repository of spatial data shared under the Open Data Commons Open Database License[43]. We observed that OSM shapefiles were substantially better in terms of coverage and accuracy compared with the World Database on Protected Areas, which is one of the most widely used databases for spatial analyses of protected areas[41]. We used OSM for both treatment and donor reserves to avoid spatial biases in our synthetic control models. We extracted OSM data using the QuickOSM plugin in QGIS v.3.22 (https://docs.3liz.org/QuickOSM/). In this study, we considered only reserves with the presence of tigers. To select the reserves with tiger presence, we intersected reserve boundaries with the International Union for Conservation of Nature global range map for *P. tigris*[44]. We obtained information on reserve establishment dates from the Environmental Information System (ENVIS) Centre on Wildlife and Protected Areas website hosted by the Wildlife Institute of India (http://wiienvis.nic.in/). We did not include tiger reserves that underwent intervention after 2015 to allow an adequate post-intervention period. Similarly, we also excluded tiger reserves that underwent the policy intervention before 2007 to allow an adequate training period for weighting the synthetic control models. We considered a total of 162 reserves in the study, out of which 45 underwent the tiger conservation intervention and the remaining 117 untreated protected areas were used as donor reserves in the synthetic control analysis (see Supplementary Table 1 for the final reserve list).

### Forest loss and spatial covariates

We used cumulative tree cover loss for each reserve between 2001 and 2020 as a response variable in our synthetic model. We compiled the cumulative tree cover loss data for each year using the Google Earth Engine platform (https://earthengine.google.com/) and Forest Cover Change dataset[45]. We collected human population density, precipitation, elevation, slope, aspect, aboveground biomass in the baseline year (2000), road length within each reserve, local purchasing power parity and minimum travel time to a city as additional covariates for matching and weighting our synthetic control models to model the underlying structural drivers of forest loss in our synthetic models. We collected the population demographic data between 2001 and 2020 from the WorldPop Global Population dataset[46]. We obtained mean precipitation in millimetres per year between 2001 and 2020 from the fifth generation of the European Centre for Medium-Range Weather Forecasts (ECMWF) Re-Analysis (ERA5) dataset[47]. We obtained geographic data for the average elevation, slope and aspect of each reserve from the Shuttle Radar Tomography Mission Digital Elevation Model in Google Earth Engine[48]. We obtained mean aboveground biomass for each reserve in 2000 as a proxy for forest intactness at the start of the study period from Global Forest Watch (https://globalforestwatch.org) for the baseline year 2000[22,49]. We obtained the road lengths within each protected area from the Socioeconomic Data and Applications Center Global Roads Open Access Data Set (gROADS) using QGIS[50]. We obtained the mean local purchasing power parity per reserve for 2000 (start of the study period) in US dollars from the Socioeconomic Data and Applications Center Global Gridded Geographically Based

Economic Data (G-Econ v.4) dataset[51]. We quantified accessibility for each reserve as the minimum travel time to a city with a population >50,000 for the year[52]. We checked for collinearity between predictor variables by determining their pairwise Pearson correlation coefficients. We did not exclude any predictor variable from the analysis because all the correlation values were <0.8 (Supplementary Fig. 2).

### Synthetic control analyses

We implemented synthetic control analysis using the tidysynth package in R[53], where we developed a synthetic counterfactual model for each reserve that underwent the tiger reserve status (*n* = 45) between 2001 and 2020. We used this method to match the cumulative forest loss of the treatment group (reserves with tiger reserve status) to a weighted model of untreated protected areas or a donor pool (protected areas with known tiger presence but not designated as tiger reserves by the NTCA, *n* = 117) before the policy intervention took place (see Supplementary Fig. 1 for an illustration of this approach). To model the underlying structural drivers of forest cover loss, we included the following reserve-level attributes associated with deforestation in protected areas as additional covariates in our matching process: human population density, road length within each reserve, precipitation, elevation, slope, aspect, aboveground biomass density in the baseline year to quantify forest intactness (2000), local purchasing power parity, age of the protected areas and minimum travel time to a city[19–24]. For tiger reserves that were an aggregate of pre-existing protected areas before the intervention, the age of the reserve was grounded on time since the establishment of the oldest constituent protected area. Matching in the pre-intervention period was grounded on the covariates listed above and the average mean cumulative forest loss before enactment of the policy[19]. We extrapolated these synthetic counterfactual models to the post-intervention period to estimate the difference between the forest loss in the synthetic reserves and observed deforestation, which represented the effect of the intervention on cumulative forest loss.

We used placebo units for significance testing. A placebo represents the synthetic model created using one of the donor units. In this case, a donor unit is treated as the treated unit and the actual treated unit is added to the donor pool. For the intervention to have an effect, that is, to reject the null hypothesis that there was no effect of the intervention on the cumulative forest loss, the differences in the performance of the donor units and the treated unit should be statistically different[18]. The ratio of the MSPE of the counterfactual unit, compared with observed values, is calculated before and after the intervention takes place. The MSPE ratios for the placebo units and the treated unit must be significantly different based on a two-sided Fisher's exact test to rule out whether the observed effect of the intervention on the outcome variable was a chance event[18] (Supplementary Table 2 and Supplementary Fig. 4). Therefore, more than 20 donor units are needed per synthetic unit to be able to yield a significance score of less than 0.05.

We separated reserves into distinct tiger landscape complexes as follows: (1) Shivalik–Gangetic (SG), (2) Northeast Hills and Brahmaputra (NEB), (3) Western Ghats (WG), (4) Central India (CI), (5) Eastern Ghats (EG) and (6) Sunderbans (SB)[54]. Donor reserves were chosen from within the same landscapes to ensure that representative untreated reserves within a comparable region were used to model synthetic controls[19]. To ensure that at least 20 donor units were available per model, we merged contiguous landscapes: (1) SG + CI + EG + SB, (2) NEB and (3) WG. Each one of these three groupings corresponded to 44, 29 and 44 donor reserves per grouping, respectively. We used non-metric multidimensional scaling (NMDS) to visualize structural differences in the covariates representing the underlying drivers of forest loss between the treatment and donor reserve groups for each of the three regions analysed. In addition, we used the analysis of similarities test to assess whether there were significant differences within clusters representing the treatment and donor pools for the three analysed regions.

Both NMDS and analysis of similarities tests were implemented using the vegan package in R[55] (see Supplementary Fig. 3 for NMDS plots).

We used the LowRankQP optimization function for fitting synthetic models given its higher accuracy and speed compared with the default ipop optimizer provided in the tidysynth package[53,56]. We used total reserve area in the synthetic control analyses to assist with model convergence. Synthetic control models were visually inspected for goodness of fit with respect to observed cumulative forest loss trajectories in the pre-intervention period. We only considered models with an unadjusted $P$ value of less than 0.05; that is, these synthetic control models performed better than at least 95% of placebo reserves created using protected areas from the donor pool in terms of the MSPE ratios described above and were retained for further analyses to estimate averted forest loss and avoided emissions. We calculated the averted forest loss per reserve as the difference between the cumulative forest loss value of the synthetic control unit and the observed loss for 2020.

## Robustness checks

We used additional robustness checks to verify the results of our synthetic control analyses. First, to account for potential anticipation effects, we split the pre-intervention period into a training and testing period[57]. For reserves that exhibited significant results (unadjusted $P < 0.05$), we simulated a backdated hypothetical intervention occurring in 2005, which is the year when the NTCA was constituted. Reserves that also exhibited significant effects from this pseudo-intervention were excluded from the analyses due to the influence of potential anticipation effects. Second, as tiger reserves tended to be larger than donor reserves, we used area-based trimming of the donor pool to assess the robustness of our results[19]. We used two thresholds: first, we ensured that donor reserves were at least a tenth of the size of a tiger reserve to be included in the donor pool for modelling the synthetic counterfactual; second, we used a more conservative threshold, where a donor reserve must be at least a quarter of the size of a tiger reserve to be included in the donor pool. We evaluated the direction, significance and magnitude of our modelled reserves with these adjusted donor pools to check for robustness of our final results. We defined direction as whether the effect of the intervention on avoided forest loss had the same sign as that observed in the untrimmed donor pool. We assessed magnitude using the condition that the effect of the intervention on avoided forest loss was within 20% of the values observed in the untrimmed donor pool. Significance testing was grounded on the placebo tests described above. As at least 20 donor units are required to obtain significance values of less than 0.05, we produced counterfactuals for 8 of the 15 tiger reserves described in our main results for the 25% threshold. Finally, to compare the quality of model fit between different groups of treated reserves, we used a bootstrap hypothesis testing approach for two independent samples, with replacement, using 9,999 iterations[58] to assess whether there were significant differences between the pre-intervention mean square error between the two groups.

## Estimating avoided emissions

To translate averted forest loss into avoided emissions, we collected mean aboveground carbon and belowground biomass carbon densities and mean uncertainty estimates for each reserve[25]. We scaled biomass density values to the intervention year using the difference between forest coverage in 2010 and the intervention year because the dataset we used represents carbon biomass estimates for 2010 (ref. 25). These values were multiplied by the averted forest loss in hectares for each reserve to obtain the mean, minimum and maximum (based on uncertainty estimates in ref. 25) total aboveground biomass in tons per reserve. We used a 10-year linear decay rate for the estimation of belowground carbon pools in forests[10]. We did not assume a hypothetical future land-use scenario in lost forest areas owing to the high degree of uncertainty associated with land-use change predictions. We then converted the net averted forest biomass value into a $tCO_2e$ using a standard Intergovernmental Panel on Climate Change emissions factor of 3.67 (ref. 10). Avoided emissions were interpreted in terms of ecosystem services through the avoided social cost of carbon due to the emissions avoided from the conservation intervention. We valued these services based on a social cost of carbon estimate of US$86 per $tCO_2e$ for India to obtain a present-day value for avoided emissions[26]. The social cost of carbon represents the economic damage from an extra ton of carbon emissions and is a commonly used measure for estimating the avoided economic damages of climate mitigation strategies[26]. Additionally, we based the estimates of the potential carbon offset value of avoided emissions on a market estimate of US$5.8 per $tCO_2e$ used in pre-existing literature[6]. This value represents what a ton of carbon offsets or negative emissions are traded at in the voluntary carbon market[10]. A carbon offset generated from an emissions reduction initiative, subject to a set of conditions, can be purchased and retired by a buyer to counterbalance their climate impact[27].

## Inclusion and ethics statement

Our study includes multiple authors from the region in which this study is based. These authors were instrumental in the study design, implementation, data analyses and manuscript preparation. In addition, we have referred to locally relevant research extensively while preparing our manuscript.

## Reporting summary

Further information on research design is available in the Nature Portfolio Reporting Summary linked to this article.

## Data availability

Our study relies on publicly available datasets for forest loss and protected area covariates. Protected area boundaries were collected using OpenStreetMap, which is shared under the Open Data Commons Open Database License (https://www.openstreetmap.org/copyright). All data compiled from the study are available in the Zenodo repository (https://doi.org/10.5281/zenodo.7711520).

## Code availability

All code used in the study is available in the Zenodo repository (https://doi.org/10.5281/zenodo.7711520).

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

## Acknowledgements

We thank F. W. S. Leong for her contributions to improving this paper. A.L. is grateful to J. S. H. Lee for her valuable guidance and feedback. L.P.K. is supported by the National Research Foundation (NRF) Singapore under its NRF Returning Singaporean Scientists Scheme (NRF-RSS2019-007). A.L. is supported by a National University of Singapore Graduate Research Scholarship. The geographical data used in our study have been made available by OpenStreetMap contributors under the Open Database License.

## Author contributions

A.L. conceived the study. A.L. designed the study with critical input from H.C.T., R.S., Y.Z., L.R.C. and L.P.K. Data were compiled by A.L., H.C.T. and R.S. A.L., R.S., Y.Z. and H.C.T. analysed the data. A.L., H.C.T., R.S., Y.Z., L.R.C. and L.P.K. contributed to writing the paper and refining drafts.

## Competing interests

The authors declare no competing interests.

## Additional information

**Extended data** is available for this paper at https://doi.org/10.1038/s41559-023-02069-x.

**Correspondence and requests for materials** should be addressed to Aakash Lamba or Lian Pin Koh.

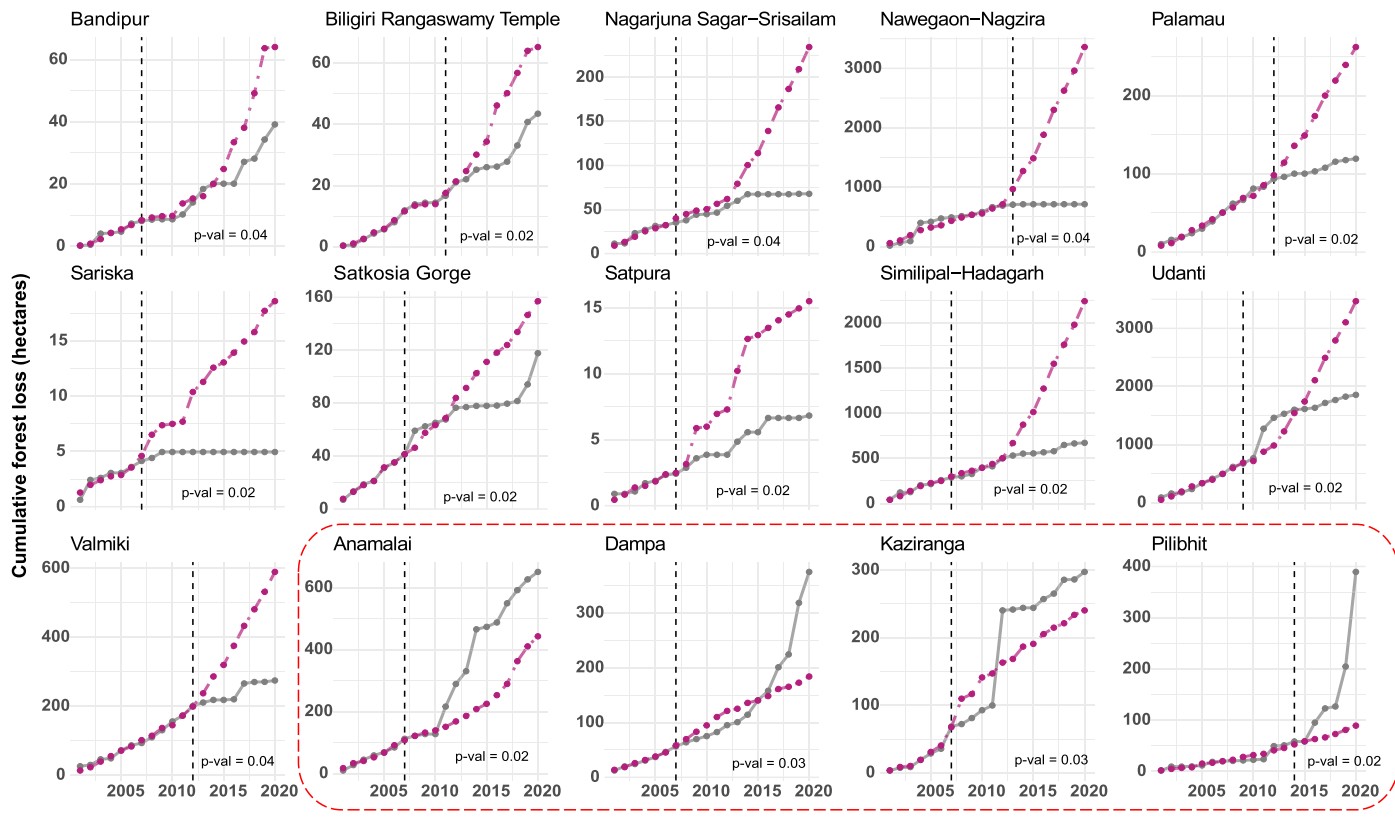

**Extended Data Fig. 1 | Trend lines for cumulative forest loss in Tiger Reserves where the conservation policy exhibited significant effects on deforestation.** Cumulative forest loss for Tiger Reserves that exhibited significant results (unadjusted p-value < 0.05) based on a two-sided Fisher's exact test to compare the ratios of pre-intervention and post-intervention mean squared prediction errors between treated reserves and placebo units (see Methods section for more details). Significance levels from placebo testing have been reported for each synthetic counterfactual in the displayed plots. Overall, 15 out of the 45 reserves exhibited significant effects after Tiger Reserves with anticipation effects ruled out (See Supplementary Figs. 5, 6 and Supplementary Table 3). Of these 15 Tiger Reserves, 11 demonstrated avoided deforestation. The remaining four reserves demonstrated higher than anticipated forest loss (highlighted in the dashed red box). The dotted pink line represents the cumulative forest loss for the synthetic control model while the dotted grey line represents observed deforestation in hectares. The vertical dashed line represents the year of implementation of the enhanced conservation policy. For each of these reserves, the synthetic control line closely tracks the observed cumulative forest loss values before the intervention.

# Reporting Summary

## Statistics

For all statistical analyses, confirm that the following items are present in the figure legend, table legend, main text, or Methods section.

| n/a | Confirmed | |
|---|---|---|
| ☐ | ☒ | The exact sample size (*n*) for each experimental group/condition, given as a discrete number and unit of measurement |
| ☒ | ☐ | A statement on whether measurements were taken from distinct samples or whether the same sample was measured repeatedly |
| ☐ | ☒ | The statistical test(s) used AND whether they are one- or two-sided *Only common tests should be described solely by name; describe more complex techniques in the Methods section.* |
| ☐ | ☒ | A description of all covariates tested |
| ☒ | ☐ | A description of any assumptions or corrections, such as tests of normality and adjustment for multiple comparisons |
| ☐ | ☒ | A full description of the statistical parameters including central tendency (e.g. means) or other basic estimates (e.g. regression coefficient) AND variation (e.g. standard deviation) or associated estimates of uncertainty (e.g. confidence intervals) |
| ☐ | ☒ | For null hypothesis testing, the test statistic (e.g. *F*, *t*, *r*) with confidence intervals, effect sizes, degrees of freedom and *P* value noted *Give P values as exact values whenever suitable.* |
| ☒ | ☐ | For Bayesian analysis, information on the choice of priors and Markov chain Monte Carlo settings |
| ☒ | ☐ | For hierarchical and complex designs, identification of the appropriate level for tests and full reporting of outcomes |
| ☐ | ☒ | Estimates of effect sizes (e.g. Cohen's *d*, Pearson's *r*), indicating how they were calculated |

*Our web collection on statistics for biologists contains articles on many of the points above.*

## Software and code

Policy information about availability of computer code

| Data collection | All data used in the analyses were based on published studies or publicly available datasets, which have been listed in the methods section. Cumulative forest loss and spatial covariates were collected using Google Earth Engine (code included). QGIS ver. 3.22 was used to extract road length per reserve. |
|---|---|
| Data analysis | All data analyses were conducted in R version 4.2.0. Synthetic control models were generated using the 'tidysynth' package (ver. 0.1.0) in R. |

For manuscripts utilizing custom algorithms or software that are central to the research but not yet described in published literature, software must be made available to editors and reviewers. We strongly encourage code deposition in a community repository (e.g. GitHub). See the Nature Portfolio guidelines for submitting code & software for further information.

## Data

Policy information about availability of data

All manuscripts must include a data availability statement. This statement should provide the following information, where applicable:
- Accession codes, unique identifiers, or web links for publicly available datasets
- A description of any restrictions on data availability
- For clinical datasets or third party data, please ensure that the statement adheres to our policy

All data generated generated from the study can be found at https://doi.org/10.5281/zenodo.7711520

# Human research participants

Policy information about studies involving human research participants and Sex and Gender in Research.

| Reporting on sex and gender | na |
| Population characteristics | na |
| Recruitment | na |
| Ethics oversight | na |

Note that full information on the approval of the study protocol must also be provided in the manuscript.

# Field-specific reporting

Please select the one below that is the best fit for your research. If you are not sure, read the appropriate sections before making your selection.

☐ Life sciences ☐ Behavioural & social sciences ☒ Ecological, evolutionary & environmental sciences

For a reference copy of the document with all sections, see nature.com/documents/nr-reporting-summary-flat.pdf

# Ecological, evolutionary & environmental sciences study design

All studies must disclose on these points even when the disclosure is negative.

| Study description | We used a synthetic controls approach to model the effects of an enhanced tiger conservation policy intervention on reducing deforestation rates in protected areas in India. We split protected areas that fall within the IUCN tiger range map into a treatment group which underwent the conservation policy (Tiger Reserves) and a 'donor pool' of untreated protected areas that did not undergo the conservation intervention. We used a weighted model of the donor pool reserves to create a synthetic counterfactual that simulated forest loss rates in treated Tiger Reserves before the policy intervention took place. These synthetic models were extrapolated into the post-intervention period, where the differences between observed cumulative forest loss in Tiger Reserves and their corresponding synthetic counterfactuals were used to estimate avoided forest loss. We translated this avoided forest loss into a carbon emissions equivalent value based on the mean above-ground and below-ground carbon biomass density in these reserves. Finally, we estimated the ecosystem services values from the avoided social cost of carbon emissions and potential carbon offset revenues from avoided deforestation due to the enhanced protection of Tiger Reserves in India. |
| Research sample | Our research sample consisted of protected areas in mainland India that fall within the global tiger range map. For each of these reserves, the outcome variable i.e. cumulative forest loss and associated spatial co-variates were collected for each year between 2001 and 2020. The observed forest loss values were obtained from the Hansen Global Forest Change v1.9 dataset. |
| Sampling strategy | Protected areas were split into those that (1) underwent an enhanced tiger conservation policy and (2) protected areas that did not undergo this policy and were thus used to create a counterfactual model to simulate cumulative forest loss in (1) had the intervention not taken place. |
| Data collection | Our analyses relied on using publicly available spatial datasets which were collected and processed using Google Earth Engine and QGIS ver. 3.22. The full list of datasets used has been described in the methods section. |
| Timing and spatial scale | Our study includes data between 2000 and 2020, with 2001 representing the first year of forest loss. The response variable i.e. cumulative forest loss and associated spatial co-variates were reported annually per reserve between 2001 and 2020. Data were included for all protected areas in mainland India that fall within the IUCN tiger range map. |
| Data exclusions | We excluded protected areas that underwent the conservation intervention less than five years after the starting period of the study in 2020 and less than five years before the end of our study period in 2020. This ensured sufficient data in the pre-intervention period to allow robust model fitting and sufficient time in the post-intervention period to observe the effects of enhanced protection. |
| Reproducibility | We have provided the data and code used in our study to ensure reproducibility of our results. We ran our counterfactual simulations multiple times to ensure that results were repeatable. Additionally, we ran a series of robustness checks to validate our results, which included tests for anticipation effects and examining the robustness of the results to trimming control units in the donor pool. |
| Randomization | Randomization was not relevant to our study since we rely on comparing observational data with synthetically produced counterfactuals to produce our results. |
| Blinding | Blinding was not relevant to the study since it used spatial analyses that worked with historical data. |

Did the study involve field work? ☐ Yes ☒ No

# Reporting for specific materials, systems and methods

We require information from authors about some types of materials, experimental systems and methods used in many studies. Here, indicate whether each material, system or method listed is relevant to your study. If you are not sure if a list item applies to your research, read the appropriate section before selecting a response.

## Materials & experimental systems

| n/a | Involved in the study |
|-----|----------------------|
| ☒ ☐ | Antibodies |
| ☒ ☐ | Eukaryotic cell lines |
| ☒ ☐ | Palaeontology and archaeology |
| ☒ ☐ | Animals and other organisms |
| ☒ ☐ | Clinical data |
| ☒ ☐ | Dual use research of concern |

## Methods

| n/a | Involved in the study |
|-----|----------------------|
| ☒ ☐ | ChIP-seq |
| ☒ ☐ | Flow cytometry |
| ☒ ☐ | MRI-based neuroimaging |

nature portfolio | reporting summary

March 2021

3