## [Peer Review File · Nature Ecology & Evolution]

Peer Review Information

Journal: Nature Ecology & Evolution

Manuscript Title: Climate co-benefits of tiger conservation

Corresponding author name(s): Aakash Lamba, Lian Pin Koh

Editorial Notes:

Reviewer Comments & Decisions:

Decision Letter, initial version:

20th September 2022

Dear Dr Lamba,

Your manuscript entitled "Climate co-benefits of tiger conservation" has now been seen by 2 reviewers, whose comments are attached. I'm sorry for the delay in returning this decision to you; we had a third reviewer lined up but they were unable to submit their review in the end.

The reviewers have raised a number of concerns which will need to be addressed before we can offer publication in Nature Ecology & Evolution. We will therefore need to see your responses to the criticisms raised and to some editorial concerns, along with a revised manuscript, before we can reach a final decision regarding publication.

We therefore invite you to revise your manuscript taking into account all reviewer and editor comments. Please highlight all changes in the manuscript text file.

* If you have not done so already please begin to revise your manuscript so that it conforms to our Article format instructions at <http://www.nature.com/natecolevol/info/final-submission>. Refer also to any guidelines provided in this letter.

2Please use the link below to submit your revised manuscript and related files:

[REDACTED]

Nature Ecology & Evolution is committed to improving transparency in authorship. As part of our efforts in this direction, we are now requesting that all authors identified as 'corresponding author' on published papers create and link their Open Researcher and Contributor Identifier (ORCID) with their account on the Manuscript Tracking System (MTS), prior to acceptance. ORCID helps the scientific community achieve unambiguous attribution of all scholarly contributions. You can create and link your ORCID from the home page of the MTS by clicking on 'Modify my Springer Nature account'. For more information please visit www.springernature.com/orcid.

[REDACTED]

Reviewers' comments:

Reviewer #2 (Remarks to the Author):

The authors make a compelling case for the carbon benefits of biodiversity interventions, in this case, improved management of protected areas selected because of their tiger populations. The methods are appropriate and the manuscript well-written. Its contribution to the literature and policy discussions could be strengthened by addressing the following points:

1) The authors argue that "demonstrating the benefits of species conservation programs on [for] climate change mitigation targets" to "further incentivise the preservation of biodiversity, whilst achieving climate change mitigation targets" is a novel paradigm. While they are offering one of the first clear evaluations of this approach, they should acknowledge (a) related literature, such as Berzaghi et al. (2022) (<https://www.pnas.org/doi/10.1073/pnas.2120426119>), and (b) limitations, most notably that this only holds for species native to carbon-rich ecosystems. Their study also has

2implications for the potential role of protected areas in general – in addition to species conservation programs - in carbon markets and NDCs.

2) The authors refer several times to the “fungible nature” of carbon offsets. They should expand on what they mean and its implications.

3) I am intrigued by the comment that “that addressing human-wildlife conflict may have potentially been a more important determinant of tiger numbers in Pilibhit as compared to the extent of available habitat.” This seems worth further exploring, e.g. comparing levels of human-wildlife conflict across the protected areas where there was a positive impact, no impact, and negative impact.

4) Check sentence structure in lines 286 – 291: “The Tiger Reserves are already protected, prior to the conservation policy, and is unlikely to qualify, and therefore are ineligible, for receiving funds from carbon markets. This implies that the USD 5.6 million is likely to remain untapped through existing avoided emissions methodologies. Instead, a possibility could be to recognize the potential for improved management of protected areas to provide additional carbon benefits an enhanced form of safeguards.”

5) The conversion to carbon emissions seems to ignore two offsetting factors: below-ground biomass of forest, and carbon stock of land use that replaces forest. Is that correct?

6) For the SCM, the donor pool is “protected areas with known tiger presence but not designated as Tiger Reserves by the NTCA.” This begs the question of why these protected areas were not designated, including whether there were structural differences.

7) In Table S1, the protected areas (treated and donor pool) should be grouped by region ((a) SG+CI+EG+SB, (b) NEB and (c) WG), since that may affect the quality of inference for each reserve. Further, it would be good to report the size of each reserve, and/or summary statistics comparing the size distribution of the treated group and donor pool in each region.

8) From Figure 1, it is difficult to discern the distance between protected areas. This is important because of the possibility of leakage or spillover effects between proximate areas.

9) Of the 27 protected areas for which no significant impact was identified, did any have poor quality synthetic controls? Or is the claim that synthetic controls closely matched treated outcomes in the pre-treatment period for all 46 protected areas?

10) Could the authors offer further detail and interpretation of the results in NE India? Three out of five protected areas where the NTCA seemed to increase forest loss are located in this region. How about the remaining protected areas in this region - in how many did NCTA prevent forest loss? What was the quality of the synthetic control matching? The comment that the finding of a negative impact of NCTA “aligns with previous literature that found that this region experienced significantly higher forest loss as compared with other parts of the country” does not really make sense if the donor pool is also from this same region.

11) The authors should provide more careful motivation of their covariates. They comment that “the method also allows the inclusion of additional covariates in the matching process to better characterise the treatment group,” but I would argue that it is not just that additional covariates are allowed, but rather that covariates representing the underlying structural drivers of forest cover loss are critical to accurately constructing the counterfactual. They seem to have a large enough donor pool to obtain weights that match historical forest cover change, but the question is whether those synthetic controls provide a good estimate of the unobserved counterfactual forest cover change in the NCTA protected areas. In particular, I wonder about including burnt area as a driver of forest cover change, since it could also be a result or a mechanical part of forest cover change, rather than a predictor.

12) Lines 407 – 408 need further explanation: We only considered models with a p-value of less than

3

0.05 and were retained for further analyses to estimate averted forest loss and avoided emissions.

13) I recommend a couple other robustness checks:

a. Backdate the intervention to assess whether there were any anticipation effects that could bias the main results;

b. Compare distribution of the covariates (i.e. the structural drivers of forest cover loss) among the treated units and the synthetic controls. Especially if there are substantial differences, consider different ways to trim the donor pool and define the covariate set, and check robustness of results to those choices.

14) Finally, the rapidly evolving literature on SCM includes new methods for inference with multiple treated units. I suggest that the authors consider implementing one of these approaches, e.g. from Xu (2017), Dube and Zipperer (2015), or Abadie and L'Hour (2019).

Reviewer #3 (Remarks to the Author):

The authors present a compelling analysis that quantifies climate/ecosystem services as ancillary benefits of enhanced forest management resulting from tiger conservation in India. The broad geographic scale of the analysis and use of the synthetic controls approach to quantify effects of enhanced tiger protection on avoided deforestation and carbon sequestration are unique aspects of the manuscript.

However, the authors overstate the novelty of the biodiversity first-carbon second paradigm (e.g. L45, L48). Multiple authors have engaged in this paradigm previously.

E.g. Díaz, S., Hector, A. and Wardle, D.A., 2009. Biodiversity in forest carbon sequestration initiatives: not just a side benefit. *Current Opinion in Environmental Sustainability*, 1(1), pp.55-60. "Our main focus, therefore is not on how carbon sequestration projects can enhance biodiversity, but rather how the protection and manipulation of biodiversity in the broad sense can enhance carbon sequestration capacity in climate change mitigation projects." Diaz et al. do make the important point that there can be varying degrees of overlap between biodiversity protection and carbon sequestration, implying a need for assessing when and where the approach will yield the most benefits.

Also see Berzaghi, F., Chami, R., Cosimano, T. and Fullenkamp, C., 2022. Financing conservation by valuing carbon services produced by wild animals. *Proceedings of the National Academy of Sciences*, 119(22), p.e2120426119.

And perhaps this one... Areas of global importance for conserving terrestrial biodiversity, carbon, and water. *Nat Ecol Evol* 5, 1499–1509 (2021). <https://doi.org/10.1038/s41559-021-01528-7>

While others have written on a biodiversity first paradigm, those efforts appear to be primarily theoretical, based on modeling, and/or focused on relatively small geographic areas. The main novelty of this study (and probably what the authors should emphasize) is that it is a broad geographic scale, empirical analysis of the benefits of species-specific conservation for climate mitigation/ecosystem services. This allows conservation practitioners, policy makers, and researchers to start to calibrate

4what sort of expectations we should have regarding co-benefits in the real world (at least in this geography).

Overall, the effects of tiger conservation on forests, i.e. 68 km² of averted forest loss over 14 years (~5 km² per year, 0.07 MtCO₂e per year), seem rather small. However, this isn't entirely unexpected. In much of India, considerable deforestation has already occurred, existing forests are relatively degraded with low biomass, and the policy intervention was not optimized for avoiding deforestation and carbon loss. In this geography, forest degradation may be more extensive than deforestation (as was recently observed for the Brazilian Amazon). If that's true, then the global forest change data, while informative, may underestimate the impacts of enhanced PA management. The authors should address these issues.

The absolute impact on climate forcing from avoided CO₂ emissions are likely small and statements about substantial climate change mitigation (e.g. P9, L215) are not well supported. It may make more sense to focus more up front on how the ancillary benefits relate to conservation budgets, benefits from ecotourism, social programs for people affected by conservation activities, etc.

Overall, the methods look sound but I'm concerned about the effect of using boundaries pulled from the OSM database. Intensive land uses may abut park boundaries in many places and inaccurate placement of the boundary could over- or underestimate deforestation, thereby over- or underestimating management impacts. Given that the authors could not obtain official boundaries, and they do not provide information on the quality or accuracy of OSM boundaries, it's worth assessing and discussing how boundary uncertainty affects their conclusions.

The authors do a good job highlighting potential benefits to society resulting from tiger conservation. They should also discuss the human costs of enhanced conservation, especially those related to villagers displaced from core areas of tiger reserves. If well implemented and conducted collaboratively, village relocation may benefit certain aspects of the lives of those displaced. However, there is reason to believe that this is usually not how relocations are conducted, resulting in harm to already disadvantaged groups.

e.g. Torri, M.C., 2011. Conservation, relocation and the social consequences of conservation policies in protected areas: Case study of the Sariska Tiger Reserve, India. *Conservation and Society*, 9(1), pp.54-64.

Rangarajan, M. and Shahabuddin, G., 2006. Displacement and relocation from protected areas: Towards a biological and historical synthesis. *Conservation and society*, 4(3), pp.359-378.

Line Edits/Comments

P3, L89 – It's unclear to me whether average biomass densities are for 2000 or for 2010. How do the authors align biomass with changes to reserve designation and biomass losses to deforestation? I see now that some of this info is in the methods but it may be worth a line describing these details in the main body of the paper. P13, L360 indicates biomass was for 2010.

Figure 1. Need an inset map. Label north arrow.

Table 1. Use units of thousands of dollars, thousands of tons, etc. to reduce the number of digits shown

P8 – The paragraph starting on L195 begs the question “Why did the conservation strategies described in the previous paragraph that worked in the successful reserves not work in these reserves?”.

P10, L242-244 – Subject-verb agreement

P10, L248 – It’s not clear to me from the numbers presented here that this is a substantial proportion of the financial costs of reserves. ~5 million USD over 14 years compared to an annual budget of 27 million USD seems like a small proportion.

*****END*****

Author Rebuttal to Initial comments

Reviewer #2 (Remarks to the Author):

The authors make a compelling case for the carbon benefits of biodiversity interventions, in this case, improved management of protected areas selected because of their tiger populations. The methods are appropriate and the manuscript well-written. Its contribution to the literature and policy discussions could be strengthened by addressing the following points:

Response: We are grateful to the reviewers for their thoughtful and constructive feedback, which has helped us significantly enhance our manuscript. We are happy to hear that the reviewers found our research question and analyses compelling. Below, we have described in detail how we have addressed individual comments from both reviewers. We have included line numbers to refer to changes in the manuscript. In addition, the revised manuscript has been attached with the changes highlighted:

1) The authors argue that “demonstrating the benefits of species conservation programs on [for] climate change mitigation targets” to “further incentivise the preservation of biodiversity, whilst achieving climate change mitigation targets” is a novel paradigm. While they are offering one of the first clear evaluations of this approach, they should acknowledge (a) related literature, such as Berzaghi et al. (2022) (<https://www.pnas.org/doi/10.1073/pnas.2120426119>), and (b) limitations, most notably that

6this only holds for species native to carbon-rich ecosystems. Their study also has implications for the potential role of protected areas in general – in addition to species conservation programs - in carbon markets and NDCs.

Response: We thank the reviewer for bringing these publications to our attention. We now acknowledge these publications which highlight a biodiversity-focussed approach to climate change mitigation. These include Berzaghi, et al. ¹, Berzaghi, et al. ², Díaz, et al. ³ and Jung, et al. ⁴. These changes are highlighted in the following lines (L43-45, L49-51 and L56-58):

“This juxtaposition of biodiversity conservation as the primary benefit instead of climate change mitigation represents an important paradigm for the preservation of natural carbon stocks⁷⁻⁹.”

“Furthermore, this paradigm potentially unlocks unforeseen opportunities for funding conservation programmes using financial instruments like carbon offsets⁸, which have been growing immensely as a source of funding for nature-based climate solutions¹⁰.”

“Revenues from the trade of fungible carbon offsets, which represent standardised and internationally tradable reductions in emissions, arising from the recognition of the climate change mitigation benefits associated with biodiversity conservation can serve as a means of closing this funding gap¹⁴”

In addition, we have added a limitations paragraph in our discussion section, which acknowledges that this paradigm only holds for species found in carbon-dense landscapes like terrestrial forests. Additionally, we discuss the important role that evaluations such as our study play in ensuring that areas of high biodiversity and carbon stock overlap are adequately funded and protected. (L298-302).

“Firstly, the carbon benefits of this approach apply primarily to species found in high-carbon ecosystems. However, given the large overlap of the world’s high biodiversity areas with carbon-rich protected areas, it is all the more urgent that such evaluations be carried out to create an evidence base for channelling more resources into the protection of such reserves⁴¹”

Finally, we discuss the avoided emissions from better management of protected areas in the context of national emissions inventories and NDCs. Although these avoided emissions are relatively small compared to the scale of the latter, it is an important contribution nonetheless, given the fact that India is one of the most highly vulnerable countries in terms of the social cost of carbon emissions (L254-259):

“Although the avoided emissions from the intervention, approximately 0.08 MtCO_{2e} per year, are small compared to India’s annual emissions and its Nationally Determined Contributions (NDCs) as part of the Paris Agreement, this is an important contribution nonetheless given the fact that India ranks as the country that is most vulnerable to the impacts of climate change in

terms of the social cost of carbon, with each additional ton of emissions leading to a loss of USD 86 to the Indian economy²⁷.”

2) The authors refer several times to the “fungible nature” of carbon offsets. They should expand on what they mean and its implications.

Response: In the context of carbon offsets, “fungibility” refers to the ability of a commodity, which in this case is a standardized unit of emissions in carbon dioxide equivalent (CO₂e) to be exchanged or traded with another good of equal quantity or value. This ability allows for these offsets to be standardised and traded internationally, and consequently allows the influx of resources/funds into biodiversity conservation (L56-58):

“Revenues from the trade of fungible carbon offsets, which represent standardised and internationally tradable reductions in emissions, arising from the recognition of the climate change mitigation benefits associated with biodiversity conservation can serve as a means of closing this funding gap¹⁴”

3) I am intrigued by the comment that “that addressing human-wildlife conflict may have potentially been a more important determinant of tiger numbers in Pilibhit as compared to the extent of available habitat.” This seems worth further exploring, e.g. comparing levels of human-wildlife conflict across the protected areas where there was a positive impact, no impact, and negative impact.

Response: The reviewer raises a very pertinent question, which we have discussed in further detail in our manuscript. Currently, there is no dataset to our knowledge that comprehensively quantifies the costs of human-tiger conflict across all the Tiger Reserves in India. However, there is a possibility that there may be an association since reserves like Nagarhole have historically demonstrated a high cost of human-tiger conflict based on previous literature and do not seem to have an effect on reducing deforestation rates from enhanced management based on our findings⁵. However, evidence for this dynamic remains unclear and we highlight this as an important area for further research (L242-249):

“This suggests that addressing human-wildlife conflict may have potentially been a more important determinant of tiger numbers in the Pilibhit as compared to the extent of available habitat¹⁷. It is unclear whether the prevalence of conflict is correlated with the performance of a Tiger Reserve in avoiding forest loss, but it is conceivable that there is an association since reserves that demonstrate no effect on avoiding forest loss like Nagarhole have previously reported high costs of human-tiger conflict from livestock damage and human casualties³⁶. Therefore, this should be an important priority for future investigation since to our knowledge no publicly available

dataset has comprehensive information on damages from human-tiger conflict across all the tiger reserves in India.”

4) Check sentence structure in lines 286 – 291: “The Tiger Reserves are already protected, prior to the conservation policy, and is unlikely to qualify, and therefore are ineligible, for receiving funds from carbon markets. This implies that the USD 5.6 million is likely to remain untapped through existing avoided emissions methodologies. Instead, a possibility could be to recognize the potential for improved management of protected areas to provide additional carbon benefits an enhanced form of safeguards.”

Response: We have improved the sentence structure in these lines (L283-289) The revised sentence has been reproduced below:

“Tiger Reserves are already protected prior to the enhanced conservation policy, and therefore are technically ineligible for receiving funds from carbon markets as they do not meet the fundamental criterion of additionality. This implies that the USD 6.2 million is likely to remain untapped through existing avoided emissions methodologies. Instead, a possibility could be to recognize the potential for improved management of protected areas to provide additional carbon benefits.”

5) The conversion to carbon emissions seems to ignore two offsetting factors: below-ground biomass of forest, and carbon stock of land use that replaces forest. Is that correct?

Response: We only considered above-ground carbon pools in our previous analyses. We have enhanced our avoided carbon emissions estimates to include below-ground forest carbon stock. We sourced the mean below-ground biomass and uncertainty values from Spawn, et al. ⁶ (see Table S2). We assume a standard 10-year linear decay rate following Koh, et al. ⁷ to model the decay of below-ground biomass. However, we did not assume a future land-use type for forest loss pixels since the synthetic controls unit for a reserve represents a hypothetical scenario, where predicting a replacement land-use type is highly uncertain given the high degree of variability in land-use predictions for forest cover in this geography⁸. These changes are now reflected in our revised methodology for emissions estimates has been described in detail in L442-451. All plots and tables have been revised to net emissions avoided that include both AGBC and BGBC estimates:

“To translate averted forest loss into avoided emissions, we collected mean above-ground carbon (AGBC) and below-ground biomass carbon (BGBC) densities and mean uncertainty estimates for each reserve²⁶. We scaled biomass density values to the intervention year using the difference between forest coverage in the year 2010 and the intervention year since the dataset used represents carbon biomass estimates for the year 2010²⁶. These values were multiplied by the averted forest loss in hectares for each reserve to obtain the mean, minimum and maximum

(based on uncertainty estimates in ²⁶) total above-ground biomass in tons per reserve. We used a 10-year linear decay rate for the estimation of below-ground carbon pools in forests¹⁰. We did not assume a hypothetical future land-use scenario in lost forest areas owing to the high degree of uncertainty associated with land-use change predictions.”

6) For the SCM, the donor pool is “protected areas with known tiger presence but not designated as Tiger Reserves by the NTCA.” This begs the question of why these protected areas were not designated, including whether there were structural differences.

Response: The designation of protected areas with tiger presence to Tiger Reserve status by the NTCA is an ongoing process with additional reserves being continually added. For instance, three new reserves have been added since 2020 ⁹. Therefore, it is likely that based on socio-political and economic developments in the future, additional reserves from the donor pools will likely be notified as Tiger Reserves in the subsequent years. To examine if there are structural differences between the treatment and donor reserves between 2000 and 2020, we utilized a dimensionality reduction approach implemented using Non-metric Multidimensional Scaling (NMDS). We (1) visually assessed any differences between the distribution of reserve-level co-variables within the two groups for each of the three geographical regions analysed and (2) used analysis of similarity or the ANOSIM test to assess whether there were significant differences between the combined distribution in co-variables for the treatment and donor pool. Both of these were implemented using the ‘vegan’ package in R ¹⁰. The spread of donor and treatment reserves was comparable for each of the three geographical regions modelled suggesting no significant difference in the distribution of the drivers of forest loss. Additionally, the ANOSIM test yielded non-significant p-values (>0.1) for comparisons between the donor and treatment pool co-variables for all three geographical regions analysed (see Fig. S3). Our methods for this approach have been described in detail in L399-407:

“To ensure that at least 20 donor units were available per model, we merged contiguous landscapes: (a) SG+CI+EG+SB, (b) NEB and (c) WG. Each one of these three groupings corresponded to 44, 29 and 44 donor reserves per grouping respectively. We used Non-metric Multidimensional Scaling (NMDS) to visualize structural differences in the covariates representing the underlying drivers of forest loss between the treatment and donor reserves groups for each of the three regions analysed. Additionally, we used the ANOSIM test to assess whether there were significant differences within clusters representing the treatment and donor pools for the three analysed regions. Both NMDS and ANOSIM were implemented using the ‘vegan’ package in R⁵⁵ (see Fig. S3 for NMDS plots).”

7) In Table S1, the protected areas (treated and donor pool) should be grouped by region ((a) SG+CI+EG+SB, (b) NEB and (c) WG), since that may affect the quality of inference for each reserve.

Further, it would be good to report the size of each reserve, and/or summary statistics comparing the size distribution of the treated group and donor pool in each region

Response: We have integrated these suggestions into our manuscript. Reserves have been grouped by regions in Table S1. Additionally, we have reported the size of each of the reserves from the Open Street Maps spatial dataset in column 4 of Table S1. Furthermore, we also acknowledge that area estimates may vary given that our shapefiles have been produced using Open Street Maps in our discussion section (L302-306):

“Second, we relied on protected area boundaries obtained from OpenStreetMaps (OSM), which may introduce uncertainties in our analyses given that these are based on publicly available user-generated data. This is, to our knowledge, the most updated and representative publicly available dataset—for instance, the most recent version of the World Database on Protected Areas (WDPA) includes shapefiles of less than 5% of India’s published reserves⁴²”

8) From Figure 1, it is difficult to discern the distance between protected areas. This is important because of the possibility of leakage or spillover effects between proximate areas.

Response: We thank the reviewer for this suggestion as this has boosted the visibility of our maps in Fig. 1. Based on their feedback, we have now divided the map in Fig. 1 into three zoomed-in panels under the country-level map for each geographical zone to better visualise the distance between each reserve.

9) Of the 27 protected areas for which no significant impact was identified, did any have poor quality synthetic controls? Or is the claim that synthetic controls closely matched treated outcomes in the pre-treatment period for all 46 protected areas?

Response: Based on a visual assessment, the synthetic controls models adequately matched the pre-intervention observations for all the modelled reserves ($n = 45$, Note: Namdapha has been excluded from the analyses since the revised intervention dates from the NTCA placed the TR intervention date before 2006⁹). To compare the quality of fits between reserves that exhibited significant results versus those that did not have significant results based on placebo testing, we compared the distribution of the pre-intervention mean squared prediction errors (mspe) between these two groups. This was done to ensure that the lack of significance was not a consequence of poor fit. Since they represent different sample sizes ($n = 15$ and 30 respectively), we used a bootstrap hypothesis testing methodology based on Efron and Tibshirani¹¹ with 9999 iterations and a significance threshold of 0.05. We found no statistical difference between mspe values between reserves that showed significant results ($n = 15$) and those with insignificant results ($n = 30$). Methods and results have been discussed in L144-146 and L437-440:

“There was no statistical difference between the mean square prediction errors in the pre-intervention period between reserves that showed significant results ($n = 15$) and those with insignificant results ($n = 30$) (bootstrap hypothesis testing, p -value = 0.21) indicating a comparable quality of fit.”

“Finally, to compare the quality of model fits between different groups of treated reserves, we used a bootstrap hypothesis testing approach for two independent samples with replacement using 9999 iterations⁵⁸ to assess whether there were significant differences between the pre-intervention mean square error between the two groups.”

10) Could the authors offer further detail and interpretation of the results in NE India? Three out of five protected areas where the NCTA seemed to increase forest loss are located in this region. How about the remaining protected areas in this region - in how many did NCTA prevent forest loss? What was the quality of the synthetic control matching? The comment that the finding of a negative impact of NCTA “aligns with previous literature that found that this region experienced significantly higher forest loss as compared with other parts of the country” does not really make sense if the donor pool is also from this same region.

Response: The reviewer has raised an important concern about the interpretation of results in North-Eastern India, which we have revised in the manuscript (L221-231). Since the donor reserves are also from the same region, baseline forest loss rates should indeed take into account regional drivers. Therefore, our interpretation of these results has been revised to focus on reserve-specific drivers of forest loss that may be leading to ineffective implementation of forest protection within these reserves. This can potentially aid reserve managers in identifying local drivers that may be leading to this increased deforestation. Furthermore, we investigated whether the lack of any Tiger Reserves in North-Eastern India that avoided deforestation could be attributed to issues with model fit. We found that model fit was comparable to other zones using the bootstrap hypothesis testing approach highlighted in point 9) above, with no statistical difference in North-Eastern and reserves in other regions in pre-intervention mspe values between the two groups (bootstrap hypothesis testing, p -value = 0.49, L146-149). Therefore, the most plausible explanation is the ineffective implementation and consequently inability of the tiger conservation policy in reducing the intrinsically high rates of deforestation in reserves in the Northeast.

“However, for some Tiger Reserves, the rate of forest loss was higher than expected compared to their synthetic counterfactual. Half of these reserves (2 of 4) fall in North-Eastern India. We postulate that this may have been caused by the prevalence of reserve-specific deforestation drivers such as encroachment, shifting agricultural practices, illegal timber trade and mining, which have historically been reported in the peripheries of notable Tiger Reserves like Kaziranga

and Dampa may have contributed to the higher than expected forest loss in these reserves^{31,32}. Our study thus highlights important priority areas for managers to mitigate local threats and consequently enhance habitat protection measures in these reserves. Overall, there were no reserves in North-Eastern India that had any avoided deforestation, despite reliable model fits, which indicates that the conservation policies were plausibly inadequate in countering the intrinsically high rates of deforestation in the region²⁵.”

“Similarly, there was no statistical difference in pre-intervention mspe values in North-Eastern Tiger reserves compared to those in regions (bootstrap hypothesis testing, p-value = 0.49).”

11) The authors should provide more careful motivation of their covariates. They comment that “the method also allows the inclusion of additional covariates in the matching process to better characterise the treatment group,” but I would argue that it is not just that additional covariates are allowed, but rather that covariates representing the underlying structural drivers of forest cover loss are critical to accurately constructing the counterfactual. They seem to have a large enough donor pool to obtain weights that match historical forest cover change, but the question is whether those synthetic controls provide a good estimate of the unobserved counterfactual forest cover change in the NCTA protected areas. In particular, I wonder about including burnt area as a driver of forest cover change, since it could also be a result or a mechanical part of forest cover change, rather than a predictor.

Response: We have integrated these comments into our manuscript and now provide a more clear rationale with supporting literature for the co-variates used in our analyses, which as the reviewer correctly pointed out were used to model the underlying structural drivers of forest loss in protected areas (L86-90, L347-350 and L372-377):

“To model the underlying structural drivers of forest cover loss we included reserve-level variables associated with deforestation, which included anthropogenic disturbances, history of protection, poverty indices, geographical attributes, forest quality at the start of the study period and climatic variables, as co-variates in the matching process (see Methods for details on covariates for drivers of forest loss)²⁰⁻²⁵.”

“We collected human population density, precipitation, elevation, slope, aspect, above-ground biomass in the baseline year (2000), road length within each reserve, local purchasing power parity and minimum travel time to a city as additional covariates for matching and weighting our synthetic controls models to model the underlying structural drivers of forest loss in our synthetic models.”

“To model the underlying structural drivers of forest cover loss, we included the following reserve-level attributes associated with deforestation in protected areas as additional covariates in our matching process: human population density, road length within each reserve, precipitation, elevation, slope, aspect, above-ground biomass density in the baseline year to quantify forest intactness (2000), local purchasing power parity, age of the protected areas and minimum travel time to a city²⁰⁻²⁵.”

Additionally, we are grateful to the reviewer for their recommendation for reconsidering burnt areas as a driver for forest loss in our analyses. As they suggested, fire may be a consequence or a mechanical part of forest cover change in this geography. Fire regimes could be dependent on climatic drivers, poverty indicators, human population densities and forest types¹²⁻¹⁴, all of which had already been included in the matching process by including the aforementioned variables in our counterfactual models. Therefore, we have excluded fire from our synthetic controls analyses.

12) Lines 407 – 408 need further explanation: We only considered models with a p-value of less than 0.05 and were retained for further analyses to estimate averted forest loss and avoided emissions.

Response: We have explained the basis for our significance testing using placebo tests in more detail (L412-415):

“We only considered models with a p-value of less than 0.05 i.e. these synthetic controls models performed better than at least 95% of placebo reserves created using protected areas from the donor pool in terms of the mspe ratios described above and were retained for further analyses to estimate averted forest loss and avoided emissions.”

13) I recommend a couple other robustness checks:

- a. Backdate the intervention to assess whether there were any anticipation effects that could bias the main results;
- b. Compare distribution of the covariates (i.e. the structural drivers of forest cover loss) among the treated units and the synthetic controls. Especially if there are substantial differences, consider different ways to trim the donor pool and define the covariate set, and check robustness of results to those choices.

Response: We are thankful for the suggestions provided by the reviewer which considerably help bolster our results. Both of these approaches have been integrated into the analyses.

(a) Firstly, we divided the pre-intervention period into a training and testing period for the 16 reserves that initially demonstrated significant effects of the tiger conservation policy based on placebo tests¹⁵.

We then use a backdated hypothetical intervention in the year 2005, which is when the National Tiger Conservation Authority was constituted, to assess whether there are potential anticipation effects in any of these reserves. Reserves that exhibit significant effects (p -value < 0.05) because of this backdated intervention were excluded from the analyses (See Fig S6, S7 and Table S3). Only Pench Tiger Reserve demonstrated potential anticipation effects and was therefore excluded from the final estimate for avoided deforestation. Our methodology and results from this approach have been discussed in L420-424 and L131-133.

“Firstly, to account for potential anticipation effects we split the pre-intervention period into a training and testing period⁵⁷. For reserves that exhibited significant results (p -value < 0.05), we simulated a back-dated hypothetical intervention occurring in the year 2005, which is the year when the NTCA was constituted. Reserves that also exhibited significant effects from this pseudo-intervention were ruled out from the analyses due to the influence of potential anticipation effects.”

“Of the 45 Tiger Reserves that underwent the conservation policy intervention, 15 showed significant but mixed results (p -value < 0.05) after reserves with anticipation effects were excluded from analyses (only Pench Tiger Reserve, see Table S3, Fig. S6 and Fig. S7).”

(b) There was no significant evidence of underlying structural differences between our donor and treatment pools when the combined distribution of reserve-level covariates (see Fig S3 and response to 6) above). These variables included human population density, precipitation, elevation, slope, aspect, above-ground biomass in the baseline year, road length within each reserve, local purchasing power parity, age of the protected areas and minimum travel time to a city. However, treated reserves tended to be larger than donor reserves (see Table S1). Although the synthetic controls method, which does not require exact matching between treatment and controls, should be able to account for these differences in natural experiments such as our study¹⁶, we decided to use area-based trimming of the donor pool reserves to assess the robustness of our final results. To do so, we used two thresholds where we (1) ensured that donor reserves must be at least a tenth of the size of a Tiger Reserve and (2) ensured that a donor reserve must be at least a quarter of the size of a Tiger Reserve to be included in the donor pool. We evaluated the direction, significance and magnitude of our modelled reserves with these adjusted donor pools to check for the robustness of our final results (see Table S4). Our methodology and results from this approach have been discussed in L424-437 and L152-158:

“Second, since Tiger Reserves tended to be larger than donor reserves, we employed area-based trimming of the donor pool to assess the robustness of our results²⁰. We used two thresholds where we first ensured that donor reserves must be at least a tenth of the size of a Tiger Reserve to be included in the donor pool for modelling the synthetic counterfactual. Similarly, we used a

more conservative threshold, where a donor reserve must be at least a quarter of the size of a Tiger Reserve to be included in the donor pool. We evaluated the direction, significance and magnitude of our modelled reserves with these adjusted donor pools to check for robustness of our final results. We defined direction as whether the effect of the intervention on avoided forest loss had the same sign as that observed in the untrimmed donor pool. The magnitude was assessed using the condition that the effect of the intervention on avoided forest loss was within 20% of the values observed in the untrimmed donor pool. Significance testing was based on the placebo tests described above. Since at least 20 donor units are required to obtain significance values of less than 0.05, we could produce counterfactuals for 8 of the 15 tiger reserves described in our main results for the 25% threshold.”

“Additionally, our results were robust to area-based trimming of donor pools (see robustness checks in Methods). Effects of the intervention with trimmed donor pools exhibited the same direction for all the modelled scenarios i.e. whether the effect was consistent in terms of avoided loss or increased loss (see Table S4). Additionally, 73% (11 out of 15) and 75% (6 out of 8) of the results with a trimmed donor pool of 10% and 25% respectively, were within our $\pm 20\%$ final findings. Similarly, over 87% (13 out of 15) and 62% (5 out of 8) of the results still exhibited significant effects based on placebo tests (See Table S4).”

14) Finally, the rapidly evolving literature on SCM includes new methods for inference with multiple treated units. I suggest that the authors consider implementing one of these approaches, e.g. from Xu (2017), Dube and Zipperer (2015), or Abadie and L’Hour (2019).

Response: We acknowledge the reviewers' suggestions for exploring new SCM methodologies in our study. We did consider the option of using an approach that models multiple treatment units in our analyses. A similar approach has been employed in recent publications like Jones, et al.¹⁷ that used the `microsynth` package in R¹⁸. However, this workflow can only model a single treatment period, whereas our scenario had staggered treatment times with intervention times ranging from 2007 to 2015. Xu¹⁹, particularly the `GSynth` was considered as an alternative. However, this package does not allow the inclusion of time-invariant co-variables in the analyses. Finally, we opted to create individual inference units with disaggregated data for each Tiger Reserve to avoid interpolation biases²⁰ and provide a more intuitive approach for reserve-level management recommendations to help managers and conservation authorities to optimise their forest protection approaches.

Reviewer #3 (Remarks to the Author):

The authors present a compelling analysis that quantifies climate/ecosystem services as ancillary benefits of enhanced forest management resulting from tiger conservation in India. The broad

geographic scale of the analysis and use of the synthetic controls approach to quantify effects of enhanced tiger protection on avoided deforestation and carbon sequestration are unique aspects of the manuscript.

Response: We are glad that the reviewer finds our analysis compelling and we thank them for their insightful feedback.

(1) However, the authors overstate the novelty of the biodiversity first-carbon second paradigm (e.g. L45, L48). Multiple authors have engaged in this paradigm previously. E.g. Díaz, S., Hector, A. and Wardle, D.A., 2009. Biodiversity in forest carbon sequestration initiatives: not just a side benefit. *Current Opinion in Environmental Sustainability*, 1(1), pp.55-60. “Our main focus, therefore is not on how carbon sequestration projects can enhance biodiversity, but rather how the protection and manipulation of biodiversity in the broad sense can enhance carbon sequestration capacity in climate change mitigation projects.” Diaz et al. do make the important point that there can be varying degrees of overlap between biodiversity protection and carbon sequestration, implying a need for assessing when and where the approach will yield the most benefits.

Also see Berzaghi, F., Chami, R., Cosimano, T. and Fullenkamp, C., 2022. Financing conservation by valuing carbon services produced by wild animals. *Proceedings of the National Academy of Sciences*, 119(22), p.e2120426119.

And perhaps this one... Areas of global importance for conserving terrestrial biodiversity, carbon, and water. *Nat Ecol Evol* 5, 1499–1509 (2021). <https://doi.org/10.1038/s41559-021-01528-7>

Response: We acknowledge that a biodiversity-first approach to climate change mitigation has been previously discussed and have included references to the studies that the reviewer has brought to our attention. This was also raised by reviewer #2, and we have addressed it accordingly. We now acknowledge publications which highlight a biodiversity-first and carbon-second paradigm. These include Berzaghi, et al. ¹, Berzaghi, et al. ², Díaz, et al. ³ and Jung, et al. ⁴. These changes are highlighted in the following lines (L43-45, L49-51 and L56-58):

“This juxtaposition of biodiversity conservation as the primary benefit instead of climate change mitigation represents an important paradigm for the preservation of natural carbon stocks⁷⁻⁹.”

“Furthermore, this paradigm potentially unlocks unforeseen opportunities for funding conservation programmes using financial instruments like carbon offsets⁸, which have been growing immensely as a source of funding for nature-based climate solutions¹⁰.”

“Revenues from the trade of fungible carbon offsets, which represent standardised and internationally tradable reductions in emissions, arising from the recognition of the climate

change mitigation benefits associated with biodiversity conservation can serve as a means of closing this funding gap¹⁴

(2) While others have written on a biodiversity first paradigm, those efforts appear to be primarily theoretical, based on modeling, and/or focused on relatively small geographic areas. The main novelty of this study (and probably what the authors should emphasize) is that it is a broad geographic scale, empirical analysis of the benefits of species-specific conservation for climate mitigation/ecosystem services. This allows conservation practitioners, policy makers, and researchers to start to calibrate what sort of expectations we should have regarding co-benefits in the real world (at least in this geography).

Response: We appreciate that reviewer recognises the novelty of our findings which offer one of the first steps towards recognising the ancillary climate co-benefits of biodiversity conservation. We agree with the reviewer that this empirical data is crucial in calibrating the scale and timeframe associated with these co-benefits, which we have described in L202-204 and L268-271:

“Our findings, thus, offer empirical evidence at a broad geographical scale to support a biodiversity-first approach to climate change mitigation.”

“These estimates provide a realistic image of the scale and associated timeframes associated with the climate co-benefits of biodiversity conservation to key stakeholders like communities, researchers and policymakers.”

(3) Overall, the effects of tiger conservation on forests, i.e. 68 km² of averted forest loss over 14 years (~5 km² per year, 0.07 MtCO₂e per year), seem rather small. However, this isn't entirely unexpected. In much of India, considerable deforestation has already occurred, existing forests are relatively degraded with low biomass, and the policy intervention was not optimized for avoiding deforestation and carbon loss. In this geography, forest degradation may be more extensive than deforestation (as was recently observed for the Brazilian Amazon). If that's true, then the global forest change data, while informative, may underestimate the impacts of enhanced PA management. The authors should address these issues.

Response: We acknowledge that our revised estimates of avoided deforestation of about 4km² per year or over 0.08 of MTCO₂e per year are relatively small, especially in the context of India's national emissions and NDCs as part of the Paris Agreement (L251-259). However, we argue that this is still an important contribution given the high social cost of carbon in India. We agree that degradation is an important driver in forest carbon loss in India and that estimates for avoided forest carbon loss from the tiger conservation policy are conservative. We could not include forest degradation in our analyses since to our knowledge the only global map of forest degradation, Vancutsem, et al. ²¹, only includes tropical moist forests, which do not include a large portion of Tiger Reserves in Central India that fall under deciduous forests ²². We have included this as a limitation in our discussion section which we highlight

as an important area for future research to get a more holistic view of the climate co-benefits of biodiversity conservation (L309-313):

“A combined estimate, which included both the reserves that experienced carbon stock loss and the high-performing reserves that averted forest loss yielded approximately USD 93 million in ecosystem services from the avoided social costs of emissions. Although the avoided emissions from the intervention, approximately 0.08 MtCO₂e per year, are small compared to India’s annual emissions and its Nationally Determined Contributions (NDCs) as part of the Paris Agreement, this is an important contribution nonetheless given the fact that India ranks as the country that is most vulnerable to the impacts of climate change in terms of the social cost of carbon, with each additional ton of emissions leading to a loss of USD 86 to the Indian economy²⁷.”

“Lastly, our study offers only a conservative estimate of avoided carbon emissions as we focus only on forest loss, whereas degradation may be an important driver of carbon loss in terrestrial forests⁴³. The development of a regional forest degradation dataset, which spans the varied forest types represented in India’s tiger reserves is a pressing priority for future research.”

(4) The absolute impact on climate forcing from avoided CO₂ emissions are likely small and statements about substantial climate change mitigation (e.g. P9, L215) are not well supported. It may make more sense to focus more up front on how the ancillary benefits relate to conservation budgets, benefits from ecotourism, social programs for people affected by conservation activities, etc.

Response: We have integrated these comments into our manuscript. We now take a more measured approach in terms of characterizing the scale of our avoided emissions estimates and refrain from describing the climate benefits as ‘substantial’. These changes are reflected in L199-201, L250-251 and L262-271 where we focus on how these ancillary benefits relate to costs like management budgets in protected areas. Furthermore, we agree with the reviewer that although the climate change mitigation benefits are small in comparison with measures like national NDCs, they nevertheless are an important move towards calibrating stakeholders with the likely scale and timeframes associated with ancillary climate benefits of species conservation (Please see the response to comments to point (3) above):

“Our study provides an unprecedented appraisal of the national tiger conservation policy in India, demonstrating there are important ancillary climate co-benefits of enhanced protection in Tiger Reserves.”

“However, despite the presence of reserves that experienced higher-than-expected forest loss, the designation of Tiger Reserves in India had a net positive climate benefit.”

“Although these annual returns are a fraction of the annual management costs of these reserves, they demonstrate that resources invested in biodiversity conservation can pay back in the form of economic benefits from ecosystem services. Additionally, had these enhanced protection measures been enacted in the untreated protected areas with tiger presence, an additional USD 38 million could have been gained from ecosystem services due to avoided emissions (based on a 0.21% rate of forest loss avoidance per hectare of forest cover since the baseline year across all analyzed tiger reserves). These estimates provide a realistic image of the scale and associated timeframes associated with the climate co-benefits of biodiversity conservation to key stakeholders like communities, researchers and policymakers.”

(5) Overall, the methods look sound but I’m concerned about the effect of using boundaries pulled from the OSM database. Intensive land uses may abut park boundaries in many places and inaccurate placement of the boundary could over- or underestimate deforestation, thereby over- or underestimating management impacts. Given that the authors could not obtain official boundaries, and they do not provide information on the quality or accuracy of OSM boundaries, it’s worth assessing and discussing how boundary uncertainty affects their conclusions.

Response: We have expanded on our rationale for choosing to use OpenStreetMaps (OSM) boundaries in our methods section (L326-330). To prevent spatial biases in our data, we used OSM boundaries for both treatment and donor reserves. The only other publicly available alternative to using OSM was the World Database on Protected Areas (WDPA), which is widely used in spatial studies that work with protected areas²³. However, less than 5% of India’s more than 900 protected areas are available on WDPA in the most recently updated version^{23,24}. We checked previous versions of the database for more complete records. The September 2016 version of WDPA had higher coverage of Indian protected areas with over 284 polygons within India present. However, this was not appropriate for use in our study as it was less accurate in terms of coverage and quality compared with OSM. Our curated OSM dataset on the other hand had over 606 reserve boundaries. Additionally, the 2016 WDPA dataset contained many reserve boundaries which were simply circular buffers around a centroid²³. These are highly inaccurate representations of protected areas as they can either overestimate protected area coverage if buffers created around points overlap with areas where reserves do not exist or can underestimate coverage if buffers overlap²⁵.

“We extracted protected area boundaries from OpenStreetMap, which is an open-source repository of spatial data shared under the Open Data Commons Open Database License. We observed that OSM shapefiles were significantly better in terms of coverage and accuracy as compared with the World Database on Protected Areas (WDPA), which is one of the most widely used databases for spatial analyses of protected areas⁴². We used OSM maps for both treatment and donor reserves to avoid spatial biases in our synthetic controls models”

20That said, we do acknowledge that OSM boundaries may introduce uncertainties in our analyses as these are user-generated datasets. We recognize this as a limitation (L302-309) in our discussion section. However, we also invite future analyses and collaborative research where we hope that reserve managers can use our framework with high-resolution on-ground data for more accurate estimates and consequently better protection of natural habitats.

“Second, we relied on protected area boundaries obtained from OpenStreetMaps (OSM), which may introduce uncertainties in our analyses given that these are based on publicly available user-generated data. This is, to our knowledge, the most updated and representative publicly available dataset—for instance, the most recent version of the World Database on Protected Areas (WDPA) includes shapefiles of less than 5% of India’s published reserves⁴². Moreover, our findings mark an important first step in recognising the potential scale of climate benefits of a biodiversity-first approach which we hope that protected area managers can further build on using our framework along with high-resolution on-ground data to aid more effective habitat protection.”

(6) The authors do a good job highlighting potential benefits to society resulting from tiger conservation. They should also discuss the human costs of enhanced conservation, especially those related to villagers displaced from core areas of tiger reserves. If well implemented and conducted collaboratively, village relocation may benefit certain aspects of the lives of those displaced. However, there is reason to believe that this is usually not how relocations are conducted, resulting in harm to already disadvantaged groups.

e.g. Torri, M.C., 2011. Conservation, relocation and the social consequences of conservation policies in protected areas: Case study of the Sariska Tiger Reserve, India. *Conservation and Society*, 9(1), pp.54-64.

Rangarajan, M. and Shahabuddin, G., 2006. Displacement and relocation from protected areas: Towards a biological and historical synthesis. *Conservation and society*, 4(3), pp.359-378.

Response: We completely agree with the reviewer that the costs of conservation borne by local communities, particularly due to relocations in and around protected areas have historically been a significant issue. We thank the reviewer for bringing these publications to our attention which we have included in the revised manuscript. In addition to highlighting these concerns about the inclusion of local communities into conservation policies, especially in terms of benefit sharing from carbon revenues, we highlight the crucial role that communities play in ensuring that conservation is implemented effectively and fairly (L293-297):

“However, given the historical costs of conservation policies borne by local communities around tiger reserves, particularly through displacement^{38,39}, it is crucial that future attempts to

integrate tiger conservation policy into carbon markets be cognizant of the cultural, social and economic needs of local communities, whose participation is crucial in equitable and effective conservation⁴⁰.”

Line Edits/Comments

P3, L89 – It’s unclear to me whether average biomass densities are for 2000 or for 2010. How do the authors align biomass with changes to reserve designation and biomass losses to deforestation? I see now that some of this info is in the methods but it may be worth a line describing these details in the main body of the paper. P13, L360 indicates biomass was for 2010.

Response: The average above-ground biomass (AGB) density, which was used as a co-covariate in the synthetic controls matching process, is from the baseline year (2000)²⁶. We used this to ensure that the AGB values used in matching, which serve as a proxy for forest quality and intactness²⁷, were from the start of the pre-intervention period. The carbon density estimates used to convert avoided deforestation to averted carbon dioxide emissions were based on Spawn, et al.⁶, which is a harmonized map for 2010, which represents the period closest to the tiger conservation policies. Spawn, et al.⁶ was used for estimating average above-ground and below-ground carbon (AGBC and BGBC) densities for each Tiger Reserve that demonstrated significant effects on forest loss rates (See Table S2). We scaled carbon density values to the intervention year using the difference between forest cover in the year 2010 and the intervention year since the dataset used represents carbon biomass estimates for the year 2010. This has been clarified in the main text in L86-90 and L93-95:

To model the underlying structural drivers of forest cover loss we included reserve-level variables associated with deforestation, which included anthropogenic disturbances, history of protection, poverty indices, geographical attributes, forest quality at the start of the study period and climatic variables, as co-variables in the matching process (see Methods for details on covariates for drivers of forest loss)²⁰⁻²⁵”

“We translated the forest change due to the designation of Tiger Reserves to equivalent CO₂e emissions averted using average above-ground and below-ground biomass carbon densities for each reserve for the year the intervention was implemented²⁶.”

Figure 1. Need an inset map. Label north arrow.

Response: We have edited Fig. 1 to include a label for the North arrow. In addition, we have now divided the map in Fig. 1 into three zoomed-in panels under the country-level map for each geographical zone to better visualise the distance between each reserve.

Table 1. Use units of thousands of dollars, thousands of tons, etc. to reduce the number of digits shown

Response: These changes have been integrated into Table 1.

P8 – The paragraph starting on L195 begs the question “Why did the conservation strategies described in the previous paragraph that worked in the successful reserves not work in these reserves?”

Response: Thank you for this suggestion. We have improved our discussion section by highlighting potential reasons for these conservation strategies not working in the reserves that underperformed in terms of avoiding forest cover loss in more detail. This was also a point that Reviewer #2 (Point 10) raised. Please find our revisions reproduced below (L221-231):

“However, for some Tiger Reserves, the rate of forest loss was higher than expected compared to their synthetic counterfactual. Half of these reserves (2 of 4) fall in North-Eastern India. We postulate that this may have been caused by the prevalence of reserve-specific deforestation drivers such as encroachment, shifting agricultural practices, illegal timber trade and mining, which have historically been reported in the peripheries of notable Tiger Reserves like Kaziranga and Dampa may have contributed to the higher than expected forest loss in these reserves^{31,32}. Our study thus highlights important priority areas for managers to mitigate local threats and consequently enhance habitat protection measures in these reserves. Overall, there were no reserves in North-Eastern India that had any avoided deforestation, despite reliable model fits, which indicates that the conservation policies were plausibly inadequate in countering the intrinsically high rates of deforestation in the region²⁵”

P10, L242-244 – Subject-verb agreement

Response: Thank you for bringing this to our attention. We have corrected this in the revised manuscript.

P10, L248 – It’s not clear to me from the numbers presented here that this is a substantial proportion of the financial costs of reserves. ~5 million USD over 14 years compared to an annual budget of 27 million USD seems like a small proportion.

Response: We thank the reviewer for raising this point. We now take a more measured approach in terms of characterizing the scale of our avoided emissions estimates and refrain from describing the climate benefits as ‘substantial’. We acknowledge that an annual return of USD 7 million from the avoided social costs of carbon emissions, or about a quarter of the annual budget, is only a fraction of the annual resources allocated to management. However, this still represents an important return given India’s high vulnerability to the impacts of climate change owing to the high social cost of carbon²⁸. Most importantly, this financial return demonstrates that funds invested in biodiversity conservation

23can help pay back to society in the form of valuable ecosystem services. These revisions are reproduced (L260-265) below:

“Moreover, the budget for Project Tiger in 2020-21 was just under USD27 million based on 2020 conversion rates (www.moef.gov.in/). More than a quarter of this was paid back in over USD 7 million per year between 2007-2020 from the avoided social cost of emissions. Although these annual returns are a fraction of the annual management costs of these reserves, they demonstrate that resources invested in biodiversity conservation can pay back in the form of economic benefits from ecosystem services.”

References

- 1 Berzaghi, F., Chami, R., Cosimano, T. & Fullenkamp, C. Financing conservation by valuing carbon services produced by wild animals. *Proceedings of the National Academy of Sciences* **119**, e2120426119 (2022).
- 2 Berzaghi, F. *et al.* Value wild animals’ carbon services to fill the biodiversity financing gap. *Nature Climate Change* **12**, 598-601 (2022).
- 3 Díaz, S., Hector, A. & Wardle, D. A. Biodiversity in forest carbon sequestration initiatives: not just a side benefit. *Current Opinion in Environmental Sustainability* **1**, 55-60 (2009).
- 4 Jung, M. *et al.* Areas of global importance for conserving terrestrial biodiversity, carbon and water. *Nature Ecology & Evolution* **5**, 1499-1509 (2021).
- 5 Gulati, S., Karanth, K. K., Le, N. A. & Noack, F. Human casualties are the dominant cost of human–wildlife conflict in India. *Proceedings of the National Academy of Sciences* **118**, e1921338118 (2021).
- 6 Spawn, S. A., Sullivan, C. C., Lark, T. J. & Gibbs, H. K. Harmonized global maps of above and belowground biomass carbon density in the year 2010. *Scientific data* **7**, 1-22 (2020).
- 7 Koh, L. P., Zeng, Y., Sarira, T. V. & Siman, K. Carbon prospecting in tropical forests for climate change mitigation. *Nature communications* **12**, 1-9 (2021).
- 8 Prestele, R. *et al.* Hotspots of uncertainty in land-use and land-cover change projections: a global-scale model comparison. *Global change biology* **22**, 3967-3983 (2016).
- 9 NTCA. National Tiger Conservation Authority of India, <<https://ntca.gov.in/>> (2022)
- 10 Oksanen, J. *et al.* The vegan package. *Community ecology package* **10**, 719 (2007).
- 11 Efron, B. & Tibshirani, R. J. *An introduction to the bootstrap*. (CRC press, 1994).
- 12 Jaiswal, R. K., Mukherjee, S., Raju, K. D. & Saxena, R. Forest fire risk zone mapping from satellite imagery and GIS. *International journal of applied earth observation and geoinformation* **4**, 1-10 (2002).
- 13 Qayum, A., Ahmad, F., Arya, R. & Singh, R. K. Predictive modeling of forest fire using geospatial tools and strategic allocation of resources: eForestFire. *Stochastic Environmental Research and Risk Assessment* **34**, 2259-2275 (2020).

- 14 Kale, M. P. *et al.* Are climate extremities changing forest fire regimes in India? An analysis using MODIS fire locations during 2003–2013 and gridded climate data of India meteorological department. *Proceedings of the National Academy of Sciences, India Section A: Physical Sciences* **87**, 827-843 (2017).
- 15 West, T. A., Börner, J., Sills, E. O. & Kontoleon, A. Overstated carbon emission reductions from voluntary REDD+ projects in the Brazilian Amazon. *Proceedings of the National Academy of Sciences* **117**, 24188-24194 (2020).
- 16 Fick, S. E., Nauman, T. W., Brungard, C. C. & Duniway, M. C. Evaluating natural experiments in ecology: using synthetic controls in assessments of remotely sensed land treatments. *Ecological Applications* **31**, e02264 (2021).
- 17 Jones, I. J. *et al.* Improving rural health care reduces illegal logging and conserves carbon in a tropical forest. *Proceedings of the National Academy of Sciences* **117**, 28515-28524 (2020).
- 18 Robbins, M. W. & Davenport, S. microsynth: Synthetic Control Methods for Disaggregated and Micro-Level Data in R. *Journal of Statistical Software* **97**, 1-31 (2021).
- 19 Xu, Y., Liu, L. & Xu, M. Y. Package ‘gsynth’. R package version 1.0.9, (2018).
- 20 Abadie, A. & L’Hour, J. A penalized synthetic control estimator for disaggregated data. *Journal of the American Statistical Association* **116**, 1817-1834 (2021).
- 21 Vancutsem, C. *et al.* Long-term (1990–2019) monitoring of forest cover changes in the humid tropics. *Science Advances* **7**, eabe1603 (2021).
- 22 Jhala, Y., Gopal, R. & Qureshi, Q. Status of tigers, co-predators and prey in India by National Tiger Conservation Authority and Wildlife Institute of India. *TRO8/001 pp* **164** (2008).
- 23 UNEP-WCMC & IUCN. Protected planet: the world database on protected areas. (2022)..
- 24 Wildlife Institute of India, ENVIS Centre on Wildlife & Protected Areas. <<http://wiienvis.nic.in/>> (2022).
- 25 Visconti, P. *et al.* Effects of errors and gaps in spatial data sets on assessment of conservation progress. *Conservation biology* **27**, 1000-1010 (2013).
- 26 Harris, N. L. *et al.* Global maps of twenty-first century forest carbon fluxes. *Nature Climate Change* **11**, 234-240 (2021).
- 27 Dawson, N. *et al.* The role of Indigenous peoples and local communities in effective and equitable conservation. *Ecology and Society* **26** (2021).
- 28 Ricke, K., Drouet, L., Caldeira, K. & Tavoni, M. Country-level social cost of carbon. *Nature Climate Change* **8**, 895-900 (2018).

Decision Letter, first revision:

256th February 2023

Dear Dr. Lamba,

Thank you for submitting your revised manuscript "Climate co-benefits of tiger conservation" (NATECOLEVOL-220717064A). It has now been seen again by the original reviewers and their comments are below. The reviewers find that the paper has improved in revision, and therefore we'll be happy in principle to publish it in Nature Ecology & Evolution, pending minor revisions to satisfy the reviewers' final requests and to comply with our editorial and formatting guidelines.

[REDACTED]

Reviewer #1 (Remarks to the Author):

The authors come out with useful information promoting the climate Co benefit aspect from tiger reserves of India. It duly talks about gaps intapping these benefits through avoided emission methodologies.

Reviewer #2 (Remarks to the Author):

Thanks for your careful response.

The following article may offer some further context for explaining and highlighting the importance of your findings regarding reserves in Northeastern India:

Ghosh-Harihar, M., An, R., Athreya, R., Borthakur, U., Chanchani, P., Chetry, D., ... & Price, T. D. (2019). Protected areas and biodiversity conservation in India. *Biological Conservation*, 237, 114-124.

Reviewer #3 (Remarks to the Author):

The authors have done an excellent job addressing my concerns and have made nice improvements to

26the manuscript. It reads well and makes an important contribution to the co-benefits literature. I have no further concerns.

Our ref: NATECOLEVOL-220717064A

27th February 2023

Dear Dr. Lamba,

Thank you for your patience as we've prepared the guidelines for final submission of your Nature Ecology & Evolution manuscript, "Climate co-benefits of tiger conservation" (NATECOLEVOL-220717064A). Please carefully follow the step-by-step instructions provided in the attached file, and add a response in each row of the table to indicate the changes that you have made. Please also check and comment on any additional marked-up edits we have proposed within the text. Ensuring that each point is addressed will help to ensure that your revised manuscript can be swiftly handed over to our production team.

****We would like to start working on your revised paper, with all of the requested files and forms, as soon as possible (preferably within two weeks). Please get in contact with us immediately if you anticipate it taking more than two weeks to submit these revised files.****

In recognition of the time and expertise our reviewers provide to Nature Ecology & Evolution's editorial process, we would like to formally acknowledge their contribution to the external peer review of your manuscript entitled "Climate co-benefits of tiger conservation". For those reviewers who give their assent, we will be publishing their names alongside the published article.

Nature Ecology & Evolution offers a Transparent Peer Review option for new original research

27manuscripts submitted after December 1st, 2019. As part of this initiative, we encourage our authors to support increased transparency into the peer review process by agreeing to have the reviewer comments, author rebuttal letters, and editorial decision letters published as a Supplementary item. When you submit your final files please clearly state in your cover letter whether or not you would like to participate in this initiative. Please note that failure to state your preference will result in delays in accepting your manuscript for publication.

Cover suggestions

As you prepare your final files we encourage you to consider whether you have any images or illustrations that may be appropriate for use on the cover of Nature Ecology & Evolution.

Nature Ecology & Evolution has now transitioned to a unified Rights Collection system which will allow our Author Services team to quickly and easily collect the rights and permissions required to publish your work. Approximately 10 days after your paper is formally accepted, you will receive an email in providing you with a link to complete the grant of rights. If your paper is eligible for Open Access, our Author Services team will also be in touch regarding any additional information that may be required to arrange payment for your article.

Please note that *Nature Ecology & Evolution* is a Transformative Journal (TJ). Authors may publish their research with us through the traditional subscription access route or make their paper immediately open access through payment of an article-processing charge (APC). Authors will not be required to make a final decision about access to their article until it has been accepted. [Find out more about Transformative Journals](https://www.springernature.com/gp/open-research/transformative-journals)

Authors may need to take specific actions to achieve [compliance with funder and institutional open access mandates](https://www.springernature.com/gp/open-research/funding/policy-compliance-faqs). If your research is supported by a funder that requires immediate open access (e.g. according to [Plan S principles](https://www.springernature.com/gp/open-research/plan-s-compliance))

28then you should select the gold OA route, and we will direct you to the compliant route where possible. For authors selecting the subscription publication route, the journal's standard licensing terms will need to be accepted, including <https://www.nature.com/nature-portfolio/editorial-policies/self-archiving-and-license-to-publish>. Those licensing terms will supersede any other terms that the author or any third party may assert apply to any version of the manuscript.

For information regarding our different publishing models please see our <https://www.springernature.com/gp/open-research/transformative-journals> Transformative Journals page. If you have any questions about costs, Open Access requirements, or our legal forms, please contact ASJournals@springernature.com.

[REDACTED]

[REDACTED]

Reviewer #1:

Remarks to the Author:

The authors come out with useful information promoting the climate Co benefit aspect from tiger reserves of India. It duly talks about gaps intapping these benefits through avoided emission methodologies.

Reviewer #2:

Remarks to the Author:

Thanks for your careful response.

The following article may offer some further context for explaining and highlighting the importance of your findings regarding reserves in Northeastern India:

Ghosh-Harihar, M., An, R., Athreya, R., Borthakur, U., Chanchani, P., Chetry, D., ... & Price, T. D. (2019). Protected areas and biodiversity conservation in India. *Biological Conservation*, 237, 114-124.

Reviewer #3:

Remarks to the Author:

The authors have done an excellent job addressing my concerns and have made nice improvements to the manuscript. It reads well and makes an important contribution to the co-benefits literature. I have no further concerns.

Author Rebuttal, first revision:

Reviewer #1 (Remarks to the Author):

The authors come out with useful information promoting the climate Co benefit aspect from tiger reserves of India. It duly talks about gaps in tapping these benefits through avoided emission methodologies.

Response: We are happy to hear that the reviewer finds the results of our study important to promoting the climate co-benefits offered by tiger reserves in India. We would like to thank the reviewer for their time and expertise in reviewing our manuscript.

Reviewer #2 (Remarks to the Author):

Thanks for your careful response.

The following article may offer some further context for explaining and highlighting the importance of your findings regarding reserves in Northeastern India:

Ghosh-Harihar, M., An, R., Athreya, R., Borthakur, U., Chanchani, P., Chetry, D., ... & Price, T. D. (2019). Protected areas and biodiversity conservation in India. *Biological Conservation*, 237, 114-124.

Response: We are grateful to the reviewer for bringing this publication to our attention. It provides valuable insights into patterns of forest loss observed in reserves in North-Eastern India. We have duly acknowledged this publication in our manuscript in the following lines (L187-192):

“We postulate that this may have been caused by the prevalence of reserve-specific deforestation drivers such as encroachment, shifting agricultural practices, illegal timber trade and mining, which have historically been reported in the peripheries of notable Tiger Reserves like Kaziranga and Dampa^{30,31}. Moreover, the remoteness and lower development of reserves in North-Eastern India have likely led to less effective enforcement and a higher risk of deforestation³².”

Reviewer #3 (Remarks to the Author):

30The authors have done an excellent job addressing my concerns and have made nice improvements to the manuscript. It reads well and makes an important contribution to the co-benefits literature. I have no further concerns.

Response: We are very thankful to the reviewer for their valuable feedback during the peer-review process, which helped us greatly improve our manuscript.

Final Decision Letter:

6th April 2023

Dear Mr Lamba,

We are pleased to inform you that your Article entitled "Climate co-benefits of tiger conservation", has now been accepted for publication in Nature Ecology & Evolution.

Over the next few weeks, your paper will be copyedited to ensure that it conforms to Nature Ecology and Evolution style. Once your paper is typeset, you will receive an email with a link to choose the appropriate publishing options for your paper and our Author Services team will be in touch regarding any additional information that may be required

You will not receive your proofs until the publishing agreement has been received through our system

Due to the importance of these deadlines, we ask you please us know now whether you will be difficult to contact over the next month. If this is the case, we ask you provide us with the contact information (email, phone and fax) of someone who will be able to check the proofs on your behalf, and who will be available to address any last-minute problems . Once your paper has been scheduled for online publication, the Nature press office will be in touch to confirm the details.

Acceptance of your manuscript is conditional on all authors' agreement with our publication policies (see www.nature.com/authors/policies/index.html). In particular your manuscript must not be published elsewhere and there must be no announcement of the work to any media outlet until the publication date (the day on which it is uploaded onto our web site).

Please note that *Nature Ecology & Evolution* is a Transformative Journal (TJ). Authors may publish their research with us through the traditional subscription access route or make their paper immediately open access through payment of an article-processing charge (APC). Authors will not be required to make a final decision about access to their article until it has been accepted. ](https://www.springernature.com/gp/open-research/transformative-journals) Find out more about Transformative Journals

Authors may need to take specific actions to achieve compliance with funder and institutional open access mandates. If your research is supported by a funder that requires immediate open access (e.g. according to Plan S principles) then you should select the gold OA route, and we will direct you to the compliant route where possible. For authors selecting the subscription publication route, the journal's standard licensing terms will need to be accepted, including https://www.nature.com/nature-portfolio/editorial-policies/self-archiving-and-license-to-publish. Those licensing terms will supersede any other terms that the author or any third party may assert apply to any version of the manuscript.

An online order form for reprints of your paper is available at https://www.nature.com/reprints/author-reprints.html. All co-authors, authors' institutions and authors' funding agencies can order reprints using the form appropriate to their geographical region.

We welcome the submission of potential cover material (including a short caption of around 40 words) related to your manuscript; suggestions should be sent to Nature Ecology & Evolution as electronic files (the image should be 300 dpi at 210 x 297 mm in either TIFF or JPEG format). Please note that such pictures should be selected more for their aesthetic appeal than for their scientific content, and that colour images work better than black and white or grayscale images. Please do not try to design a cover with the Nature Ecology & Evolution logo etc., and please do not submit composites of images related to your work. I am sure you will understand that we cannot make any promise as to whether any of your suggestions might be selected for the cover of the journal.

To assist our authors in disseminating their research to the broader community, our SharedIt initiative provides you with a unique shareable link that will allow anyone (with or without a subscription) to read the published article. Recipients of the link with a subscription will also be able to download and

32print the PDF.

You can generate the link yourself when you receive your article DOI by entering it here: <http://authors.springernature.com/share>.

[REDACTED]

P.S. Click on the following link if you would like to recommend Nature Ecology & Evolution to your librarian <http://www.nature.com/subscriptions/recommend.html#forms>

** Visit the Springer Nature Editorial and Publishing website at http://editorial-jobs.springernature.com?utm_source=ejp_NEcoE_email&utm_medium=ejp_NEcoE_email&utm_campaign=ejp_NEcoE for more information about our career opportunities. If you have any questions please click [here](mailto:editorial.publishing.jobs@springernature.com). **